# NRAC controls CD36-mediated fatty acid uptake in adipocytes and lipid clearance in vivo

Inderjeet Singh[1,2,3], Yasuhiro Onogi [1,2,4], Filipe Menezes [5], Dina Khasanova[5], Lingru Kang[1,2], Chenxi Wang[1,2], Julio Ruiz-Trave [2,6], Sapna Sharma [2,7,8], Ahmed Khalil [1,2,6], Valentin K Reichenbach [1,2], Yingzi Shi[1,2], Andrew Flatley[9], Xiaocheng Yan[1,2], Andreas Israel[1,2], Nathalia R V Dragano [2,10], Juan Antonio Aguilar-Pimentel[10], Anne Hoffmann[11], Adhideb Ghosh [12], Falko Noé [12], Christian Wolfrum [12], Sebastian Cucuruz[2,13], Ann-Christine König[14], Ingo Burtscher [2,11], Stefanie M Hauck [14], Heiko Lickert[2,13], Susanna M Hofmann [2,13,15], Regina Feederle [9,16], Sonja C Schriever [2,17], Rene Hernandez-Bautista [1,2], Gencer Sancar [2,18], Alberto Cebrian-Serrano[2,6], Igor Tetko [5], Helmut Fuchs[2,10], Valérie Gailus-Durner[2,10], Matthias Blüher[2,11,19,20], Martin Hrabě de Angelis[2,10,21] & Siegfried Ussar [1,2✉]

## Abstract

Adipose tissue is a central organiser of systemic lipid homeostasis and a pharmacological target in obesity, orchestrating cellular responses to environmental cues. Nutritionally regulated adipose and cardiac enriched protein (NRAC) is a small adipocyte-specific transmembrane protein with unknown function. Here, we show that Nrac directly interacts with scavenger receptor CD36 via its first transmembrane domain. Forming a complex with CD36 and caveolin-1 under low extracellular fatty acid (FA) concentrations, NRAC modulates CD36-dependent fatty acid uptake in adipocytes. Upon increase in extracellular FA levels, NRAC is ubiquitinated and internalised, leading to CD36's dissociation from caveolin-1 and clathrin-mediated endocytosis. This results in increased fatty acid uptake into fat cells, adipocyte hypertrophy, increased fat mass and elevated lipid clearance from the blood in chow-diet-fed mice. Finally, human *NRAC* expression and the intronic SNP rs12878589 are associated with body fat distribution and obesity. Together, these findings reveal a novel regulatory mechanism by which adipocytes sense and respond to extracellular fatty acid availability to fine-tune lipid uptake and storage at cellular and organismal level.

**Keywords** CD36; Clathrin-mediated Endocytosis; Adipose Tissue; Hypertrophy; Fatty Acid Uptake
**Subject Categories** Membranes & Trafficking; Metabolism

## Introduction

Adipose tissue is centrally important for the regulation of glucose and lipid homeostasis. Too little and too much adipose mass result in the development of systemic insulin resistance and the metabolic syndrome (Bluher, 2020; Grundy, 2015; Klein et al, 2022). Thus, adipocytes continuously sense pleiotropic nervous, endocrine and nutritional/environmental inputs to adjust their endocrine and lipid handling activity. Most of these inputs are processed at the adipocyte cell membrane through various classes of transmembrane receptors and transporters (Onogi et al, 2020). Fatty acid uptake and release are among the most important functions of white adipocytes. Fatty acid uptake is initiated by the release of free fatty

[1]Research Unit Adipocytes & Metabolism (ADM), Helmholtz Diabetes Center, Helmholtz Zentrum München, German Research Center for Environmental Health GmbH, Neuherberg 85764, Germany. [2]German Center for Diabetes Research (DZD), Neuherberg 85764, Germany. [3]Department of Medicine, Technische Universität München, Munich, Germany. [4]Research Center for Pre-Disease Science (RCPDS), University of Toyama, Sugitani 2630, Toyama 9300194, Japan. [5]Molecular Targets and Therapeutics Center, Institute of Structural Biology, Neuherberg 85764, Germany. [6]Institute for Diabetes & Obesity, Helmholtz Diabetes Center, Helmholtz Center Munich, Helmholtz Zentrum München, German Research Center for Environmental Health GmbH, Ingolstaedter Landstrasse 1, Neuherberg 85764, Germany. [7]Research Unit of Molecular Epidemiology, Helmholtz Zentrum München, Neuherberg 85764, Germany. [8]Institute of Epidemiology, Helmholtz Zentrum München, Neuherberg 85764, Germany. [9]Core Facility Monoclonal Antibodies (CF-MAB), Helmholtz Zentrum München, German Research Center for Environmental Health GmbH, Neuherberg 85764, Germany. [10]Institute of Experimental Genetics, German Mouse Clinic, Helmholtz Zentrum München, German Research Center for Environmental Health GmbH, Neuherberg 85764, Germany. [11]Helmholtz Institute for Metabolic, Obesity and Vascular Research (HI-MAG), Helmholtz Zentrum München at the University of Leipzig and University Hospital Leipzig, Leipzig, Germany. [12]Institute of Food, Nutrition and Health, ETH Zurich, Schwerzenbach 8092, Switzerland. [13]Institute of Diabetes and Regeneration Research (IDR), Helmholtz Diabetes Center, Helmholtz Zentrum Munich, Neuherberg 85764, Germany. [14]Metabolomics and Proteomics Core (MPC), Helmholtz Center Munich, Neuherberg, Germany. [15]Department of Medicine IV, University Hospital, Ludwig-Maximilians University, Munich, Germany. [16]Munich Cluster for Systems Neurology (SyNergy), Feodor-Lynen-Straße 17, Munich D-81377, Germany. [17]Research Unit NeuroBiology of Diabetes, Helmholtz Munich, Ingolstädter Landstraße 1, Neuherberg 85764, Germany. [18]Department of Internal Medicine IV, Division of Diabetology, Endocrinology and Nephrology, University of Tübingen, Tübingen, Germany. [19]Institute for Diabetes Research and Metabolic Diseases, Helmholtz Center Munich, University of Tübingen, Tübingen, Germany. [20]Medical Department III – Endocrinology, Nephrology, Rheumatology, University of Leipzig Medical Center, Leipzig 04103, Germany. [21]Chair of Experimental Genetics, TUM School of Life Sciences, Technische Universität München, Freising 85354, Germany. ✉E-mail: siegfried.ussar@helmholtz-munich.de

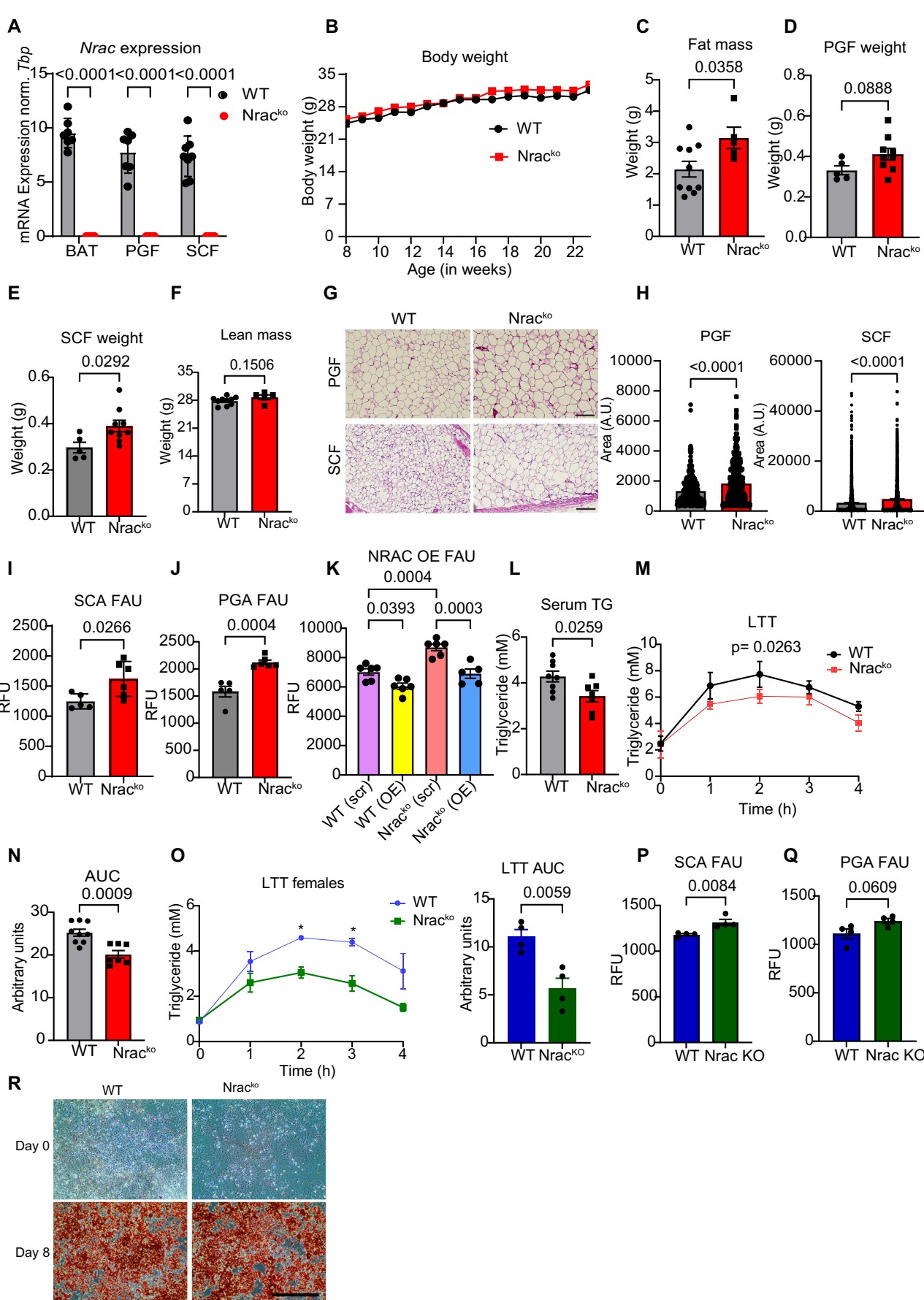

**Figure 1. NRAC deletion promotes adipocyte fatty acid uptake in chow diet-fed male mice.**

(A) mRNA expression of *Nrac* in BAT, PGF and SCF tissues (*n* = 7 wt, 5 ko). (B) Body weight (*n* = 9 wt, 5 ko). (C) Fat mass (*n* = 9 wt, 5 ko, Age 15 weeks). Tissue weight of (D) PGF and (E) SCF (*n* = 6 wt, 8 ko, age 23 weeks). (F) Lean mass (*n* = 9 wt, 5 ko, Age 15 weeks). (G) H&E staining of PGF and SCF (age 23 weeks) (scale bar 200 μm)(H) Quantification of adipocyte cell sizes. (I, J) Fatty acid uptake (FAU) in the primary isolated adipocytes from (I) SCF and (J) PGF of male mice (*n* = 5 wt, 6 ko, age 12 weeks). (K) Fatty acid uptake in stably NRAC overexpressing (OE) wt and Nrac[ko] adipocytes. Scr plasmid was used as control. (L) Serum Triglyceride (TG) levels of 23 weeks old mice (*n* = 8 wt, 7 ko). (M) Lipid tolerance test (LTT) (*n* = 9 wt, 7 ko) (age 14 weeks) and, (N) baseline corrected area under the curve (AUC), (O) Lipid tolerance test in chow diet fed female mice (*n* = 4 wt. 4ko). Fatty acid uptake in primary (P) subcutaneous adipocytes (SCA) and (Q) perigonadal adipocytes (PGA) isolated from female mice (*n* = 4 wt, 4ko). (R) Oil red O staining on the primary differentiated subcutaneous adipocytes (scale bar 200 μm). Data are represented as mean ± SEM. \**P* < 0.05, \*\**P* < 0.01, \*\*\**P* < 0.001 by Student's *t* test or one or two-way ANOVA was used. Source data are available online for this figure.

acids from triglyceride rich liporpoteins by endothelial lipoprotein lipase (LPL) (Boren et al, 2022; Gonzales and Orlando, 2007) and subsequent uptake of fatty acids into adipocytes. About 50% of all fatty acids in adipocytes are taken up by the long-chain fatty acid translocase CD36 (Goldberg et al, 2009; Hao et al, 2020). Unlike classical transporters, CD36 transports fatty acids into the adipocytes via endocytosis. Previous studies showed that CD36 endocytosis takes place mainly via caveolin-mediated endocytosis (Hao et al, 2020), albeit clathrin mediated endocytosis has been reports as well (Zeng et al, 2003). The regulation of CD36 endocytosis is important as CD36 localization in adipocytes is the major regulator of CD36-mediated fatty acid transport (Brailey et al, 2022; Daquinag et al, 2021; Hao et al, 2020; Peche et al, 2023; Wang et al, 2024). However, compared to lipolysis (Ali et al, 2015), relatively little is known about the modulation of fatty acid uptake into adipocytes, or mechanisms distinguishing CD36-mediated fatty acid uptake in adipocytes compared to other cell types such as cardiac and skeletal muscle, liver and macrophages (Ali et al, 2015). Cell type-specific regulation of CD36-mediated fatty acid uptake could be mediated by modulation of transport activity by co-receptors or other modulatory proteins with a more restricted expression pattern.

To date, very few adipocyte-selective cell surface proteins are known (Ussar et al, 2014). Among them, NRAC (Nutritionally regulated adipose and cardiac enriched protein; mouse homolog *A530016L24Rik*, human homolog- *C14ORF180*) shows some of the highest specificity to white and brown adipocytes, with additional low expression in heart (Zhang et al, 2012). NRAC is a two-pass membrane protein with both the N- and C-terminus located intracellularly and a small extracellular loop, with no additional identified protein domains. Blast searches do not reveal any homologous eukaryotic proteins that could indicate a potential function of NRAC. However, previous studies showed down-regulation of *Nrac* gene expression in white adipose tissue of fasted and obese mice, with little impact of these nutritional changes in *Nrac* expression in brown adipose tissue (BAT) (Zhang et al, 2012). Knockdown studies in cultured in vitro differentiated adipocytes suggested a role of NRAC in adipocyte differentiation or lipid accumulation (Kerr et al, 2019). However, if and what role NRAC could play in adipocyte lipid storage in vivo remains unknown. Here we show that NRAC acts as a rheostat for CD36-mediated fatty acid uptake, dependent on the extracellular fatty acid concentration. We provide data that under low extracellular fatty acid concentrations NRAC forms a complex with CD36 and caveolin-1, limiting fatty acid uptake. Upon loss of NRAC or an increase in extracellular fatty acid levels, leading to a reduction in surface localization of NRAC via increased ubiquitination resulting

in CD36 dissociation from caveolin-1 and is internalized via clathrin-mediated endocytosis. This results in increased fatty acid uptake into adipocytes and adipocyte hypertrophy, increased fat mass and elevated lipid clearance from the blood in chow diet fed mice.

## Results

### NRAC deletion leads to hypertrophic white adipocytes due to increased fatty acid uptake

NRAC is predominantly expressed in white and brown adipocytes with little additional expression reported in heart (Zhang et al, 2012). To investigate the function of NRAC in vivo, we obtained whole-body *Nrac* knockout mice and verified complete loss of *Nrac* expression by qPCR (Fig. 1A). We also developed a rat monoclonal antibody targeting the intracellular N-terminus of murine NRAC, detecting murine NRAC in NRAC-GFP transfected cells (Fig. EV1A), as well as endogenous levels by western blotting (Fig. EV1B), immunostaining (Fig. EV1C), and immunoprecipitation (Fig. EV1D).

Chow diet fed male *Nrac* knockout (Nrac[ko]) mice did not show differences in body weight (Fig. 1B). However, male Nrac[ko] mice had increased total fat mass (Fig. 1C), and subcutaneous (SCF) and perigonadal (PGF) adipose tissue mass (Fig. 1D,E), with no significant differences in lean mass (Fig. 1F) at 15 weeks of age compared to wild-type (wt) littermate controls. Histological analysis of SCF and PGF showed increased mean adipocyte size in both depots of Nrac[ko] mice (Fig. 1G,H), but no signs of increased immune cell infiltration (Fig. 1G) or elevated expression of *Tnfa* or *Il6* (Fig. EV1E–H). Weights of liver, heart and pancreas showed no genotype-specific differences (Fig. EV1I). Elevated fatty acid uptake is a major driver of adipocyte hypertrophy (Gonzales and Orlando, 2007), which was elevated in primary subcutaneous and perigonadal adipocytes from Nrac[ko] mice compared to wt littermate controls (Fig. 1I,J). Interestingly, overexpression of NRAC in wt and Nrac[ko] in vitro differentiated adipocytes led to a reduction in fatty acid uptake compared to the controls (Fig. 2K). Supporting the physiological relevance of the increased adipocyte fatty acid uptake, we found reduced random fed serum triglyceride levels (Fig. 1L) and increased lipid clearance during a lipid tolerance test (Fig. 1M,N) in Nrac[ko] mice compared to wt littermate controls. Similar to male Nrac[ko] mice, Nrac[ko] females exhibited enhanced lipid clearance (Fig. 1O) and increased fatty acid uptake in subcutaneous adipocytes (Fig. 1P), with a similar trend in perigonadal adipocytes (Fig. 1Q).

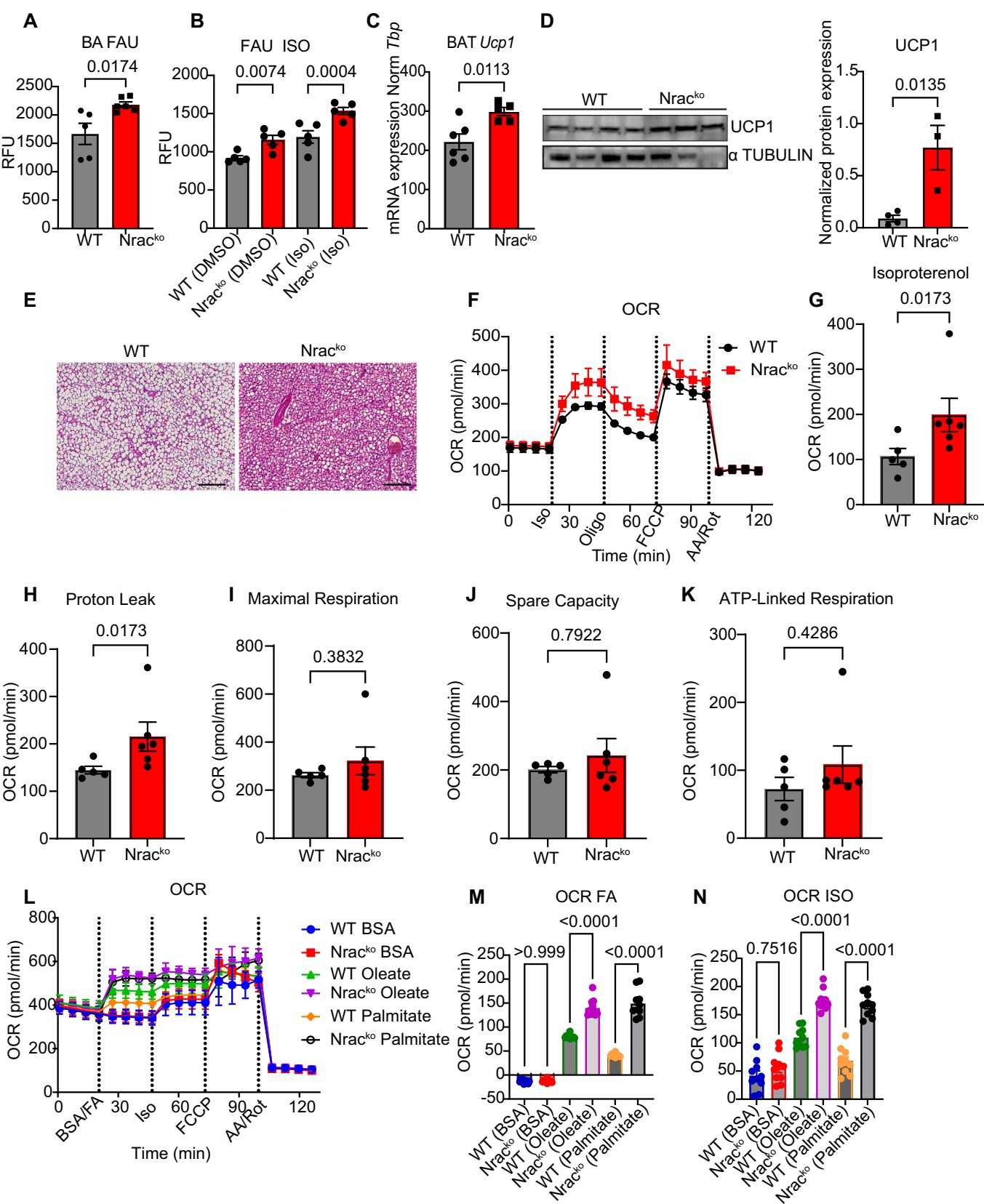

**Figure 2. NRAC deletion increases UCP1 in BAT and oxygen consumption rates in brown adipocytes.**

(A) Fatty acid uptake (FAU) in primary mature brown adipocytes (BA) ($n = 5$ wt, 6 ko, age 12 weeks). (B) FAU in differentiated brown adipocytes, treated with either DMSO or 5 µM isoproterenol for 20 min ($n = 5$). (C) *Ucp1* mRNA ($n = 6$ wt, 8 ko) and (D) protein expression ($n = 4$ wt and 3 ko) and quantification. (E) H&E staining of BAT (age 23 weeks) (scale bar 200 µm). (F) Oxygen consumption rate (OCR) in primary differentiated mature brown adipocytes (Iso Isoproterenol, Oligo Oligomycin, FCCP Carbonyl cyanide-4 (trifluoromethoxy), AA Antimycin A, Rot Rotenone), (G) OCR after isoproterenol injection (H) proton leak and (I) maximal respiratory capacity (J) spare capacity, (K) ATP-linked respiration ($n = 5$ wt, 6 ko, age 12 weeks). (L) Oxygen consumption rate in the differentiated adipocytes from wt and Nrac^ko cells. BSA conjugated oleate and palmitate (100 µM) were injected in the first port, Isoproterenol (Iso) in the second port, FCCP (Carbonyl cyanide-4 (trifluoromethoxy) in 3^rd port and AA (Antimycin A) and Rot (Rotenone) in the 4th port. (M) OCR calculated after subtracting fatty acid (FA) stimulated values from basal and (N) isoproterenol (Iso) stimulated OCR from basal ($n = 10$). Data are shown as mean ± SEM. *$P < 0.05$, **$P < 0.01$, ***$P < 0.001$ by Student's *t* test in all figures except (E) where two-way ANOVA was used. Data on figure G-K were analyzed via Mann–Mann-Whitney *U* test as data were not normally distributed. Source data are available online for this figure.

However, the observed adipose hypertrophy in chow diet fed male Nrac^ko mice did not impair systemic glucose (Fig. EV1J) or insulin metabolism (Fig. EV1K), as glucose and insulin tolerance as well as percentage of glycated haemoglobin (%HbA1c) were not different between Nrac^ko and wt littermate control mice (Fig. EV1L). Moreover, female mice did not show genotype-specific differences in body weight (Fig. EV1M), body composition (Fig. EV1N,O), glucose metabolism, (Fig. EV1P,Q), insulin sensitivity (Fig. EV1R) and adipose histology (Fig. EV1S,T). Assessment of NRAC expression between male and female mice across adipose depots revealed comparable expression between male and female adipose tissues (Fig. EV1U–W). Thus, the absence of adipose hypertrophy in female compared to male mice is most likely not the result of mechanistic differences of NRAC between male and female mice but rather a consequence of the increased whole-body energy expenditure in female mice (Mauvais-Jarvis et al, 2013).

Previous studies showed that a knockdown of NRAC impairs adipogenic differentiation in vitro (Kerr et al, 2019; Zhang et al, 2012), which could contribute to compensatory adipocyte hypertrophy (Bakker et al, 2006). We did not observe any differences in adipogenesis of subcutaneous wt and Nrac^ko preadipocytes differentiated in vitro, as shown by Oil Red O staining (Fig. 1R). Thus, loss of NRAC impacts on systemic lipid metabolism by redirecting fatty acids to white adipocytes for storage.

## NRAC deletion promotes thermogenesis in brown adipocytes without affecting systemic energy expenditure

In addition to white adipocytes, NRAC is expressed in brown adipocytes. Fatty acids induce UCP-1 mediated mitochondrial uncoupling and thermogenesis (Bartelt et al, 2011). Therefore, we investigated the brown adipose tissue (BAT) phenotype of Nrac^ko mice. Like white adipocytes, NRAC knockout increased fatty acid uptake of primary brown adipocytes (Fig. 2A). Moreover, stimulation of differentiated adipocytes with isoproterenol led to increased fatty acid uptake (Fig. 2B). In line with elevated fatty acid influx, we observed slightly increased *Ucp1* mRNA (Fig. 2C) and protein levels in Nrac^ko BAT (Fig. 2D). Histologically, we observed reduced lipid droplet size in BAT of Nrac^ko mice (Fig. 2E). To assess the effects of NRAC deletion on brown adipocyte energy metabolism, we differentiated primary brown adipocytes in vitro and measured oxygen consumption using a Seahorse Extracellular Flux analyzer. Nrac^ko brown adipocytes showed significantly increased oxygen consumption rates (OCR) upon isoproterenol stimulation, as well as increased proton leakage (Fig. 2F–H). No

differences were observed upon FCCP induced maximal respiration (Fig. 2I), spare capacity (Fig. 2J) and ATP-linked respiration (Fig. 2K) indicating that Nrac^ko brown adipocytes have higher thermogenic capacity, but require the induction of lipolysis. This is likely due to low extracellular fatty acid concentrations in the assay medium.

Previous studies have demonstrated that fatty acid stimulation enhances the thermogenic capacity of brown adipocytes (Li et al, 2014), potentially through increased internalization of CD36. To investigate whether the elevated fatty acid uptake observed in Nrac^ko adipocytes contributes to enhanced thermogenic function, we administered fatty acids (oleate and palmitate) via the Seahorse assay port. Notably, fatty acid stimulation led to a significant increase in OCR, with Nrac^ko adipocytes exhibiting a markedly higher response compared to controls (Figs. 2L–N and EV2A–C). These findings suggest that loss of Nrac potentiates thermogenic activation through enhanced fatty acid-driven respiration. To explore the physiological regulation of NRAC, we examined its cellular localization under various conditions, including starvation, acute cold exposure, and treatment with the β3-adrenergic agonist CL-316243. Among those, cold exposure and CL treatment resulted in a minor reduction in NRAC surface localization (Fig. EV2D).

Measurement of whole-body energy expenditure and substrate utilization, did not reveal any differences between Nrac^ko and wt littermate mice (Fig. EV2E–H). Moreover, acute treatment with CL-316243 (Fig. EV2I,J), or acute cold exposure (Fig. EV2K,L) also did not induce differences in systemic energy expenditure or substrate utilization. Thus, NRAC deletion increases brown adipocyte fatty acid influx and thermogenesis. However, the reduced serum triglyceride concentration appears to limit or eliminate the systemic consequences of this BAT phenotype.

## Loss of NRAC enhances adipocyte fatty acid uptake by modulating CD36 endocytic pathways

To understand the mechanism underlying the increased adipocyte fatty acid uptake upon loss of NRAC, we identified interaction partners using proteomics following NRAC immunoprecipitation. NRAC and its interaction partners were immunoprecipitated using our monoclonal rat antibody from PGF of wt mice. PGF from Nrac^ko mice was used as control. Using this approach, we identified 13 NRAC-interacting proteins (Fig. 3A). Among those, we detected proteins related to lipid uptake including CD36, caveolin-1, and cavin 1. Interaction of NRAC with CD36 and caveolin-1 was confirmed using immunoprecipitation from BAT, SCF (Fig. EV3A) and PGF (Fig. 3B) with subsequent western blotting.

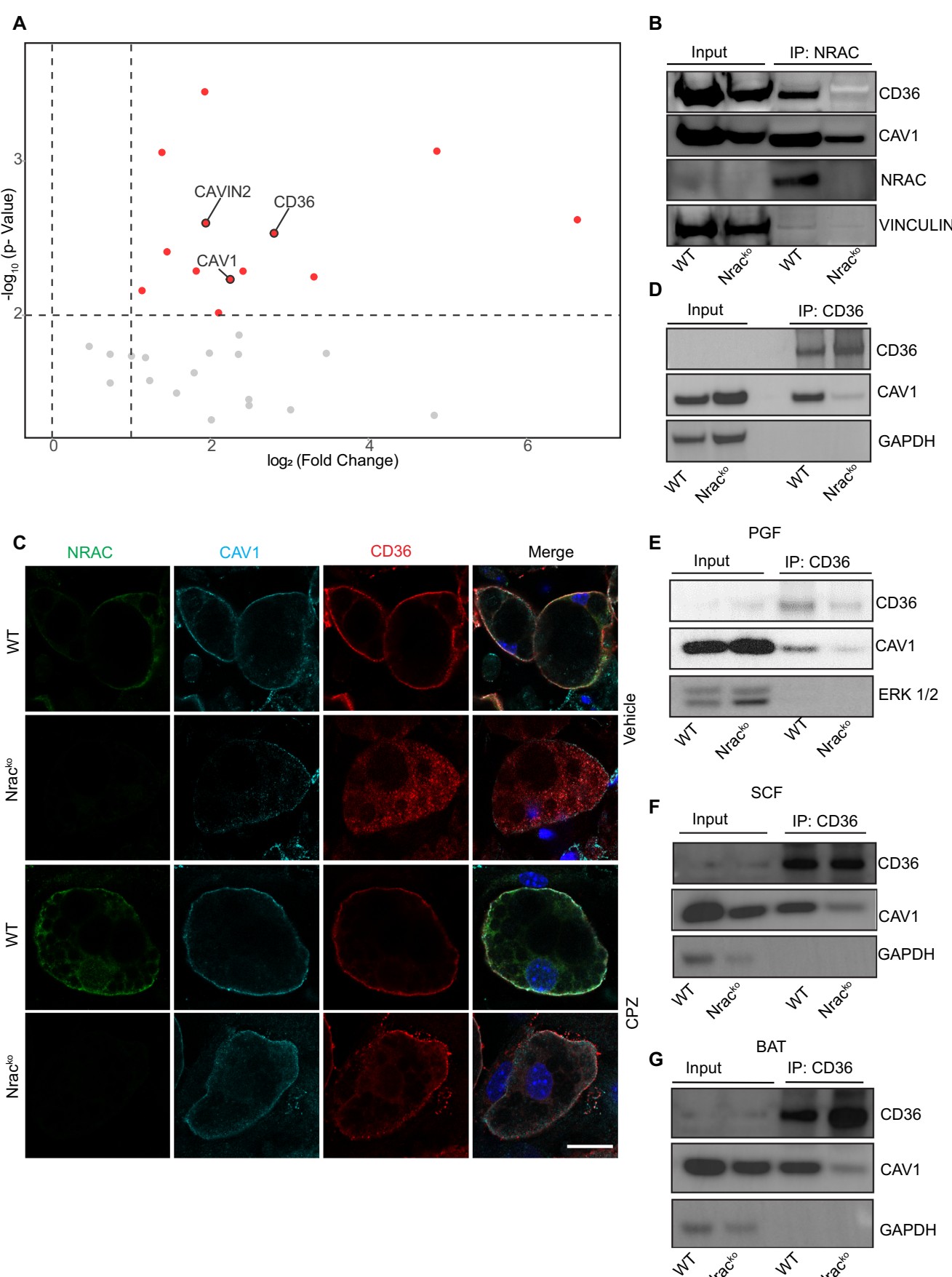

**Figure 3.   NRAC regulates CD36/ caveolin-1 interaction and CD36 internalization.**

(**A**) Mass spectrometry analysis of immunoprecipitated samples from PGF of wt and Nrac^ko mice ($n = 4$). Proteins present in at least two wt samples, a wt to ko ratio >2 and an adjusted *P* value < 0.05 were blotted. The significance threshold was set at log2 fold change above 1 and -log₁₀(*P* value) above 2. Significantly up-regulated proteins are shown in red. (**B**) Co-immunoprecipitation of NRAC with CD36 and CAVEOLIN-1 (CAV1). (**C**) Immunostaining of primary differentiated mature adipocytes either vehicle (0.05% ethanol) or Chlorpromazine (CPZ) (10 µM) treated for 1 h (scale 25 µm). (**D**) Co-immunoprecipitation of CD36 and CAV-1 from differentiated adipocytes. Co-immunoprecipitation of CD36 and CAV-1 from PGF (**E**), SCF (**F**) and BAT (**G**) of male mice. Source data are available online for this figure.

CD36 facilitates long-chain fatty acid import mainly via caveolin-mediated endocytosis (Hao et al, 2020). Using primary in vitro differentiated subcutaneous adipocytes from wt and Nrac^ko mice, we studied the spatial distribution of CD36 and caveolin-1. Immunostaining for CD36, caveolin-1 and NRAC in wild-type adipocytes showed that all proteins localized at the plasma membrane. However, while caveolin-1 was retained at the cell surface in Nrac^ko adipocytes, CD36 was no longer at the plasma membrane, but appeared in clusters accumulating around lipid droplets in primary (Fig. 3C) as well as in immortalized differentiated white adipocytes (Fig. EV3B). Immunoprecipitation of CD36 confirmed a strongly reduced interaction between CD36 and caveolin-1 in Nrac^ko adipocytes (Fig. 3D). Moreover, the interaction of CD36 and caveolin-1 was strongly reduced in PGF, SCF and BAT of Nrac^ko mice compared to wt littermate controls (Fig. 3E–G), verifying the phenotype in vivo.

CD36 transports fatty acids via endocytosis of the fatty acid/ CD36 complex. Loss of NRAC resulted in a dissociation between CD36 and caveolin-1, indicating that upon loss of NRAC, CD36 is utilizing an alternative endocytic pathway. Indeed, inhibition of clathrin-mediated endocytosis using chlorpromazine for one hour, restored CD36 cell surface localization (Figs. 3C and EV3B). Thus, loss of NRAC results in a switch from caveolin to clathrin-mediated CD36 endocytosis, increasing fatty acid uptake.

## NRAC interacts with CD36 via its first transmembrane domain

Mouse NRAC is a 165-amino-acid, two-pass membrane protein composed of a large N-terminal intracellular domain (id1, residues 1–112), two transmembrane domains (tm1, residues 113–135; tm2, residues 145–164), and an extracellular loop (el, residues 136–144). DeepLoc 2.0 (Thumuluri et al, 2022) predictions indicate that the signal peptide overlaps with the tm1 region (residues 110–130). To investigate how NRAC interacts with CD36 and to identify the domain responsible for this interaction, we generated several Nrac mutant constructs. One construct lacked tm1 (NRAC 1–112 + SP), where the signal peptide and transmembrane domain of CD36 (SP = GLIAGAVIGAVLAVFGGILMPV) was used. Additional deletion constructs included NRAC 1–135 (containing id1 and tm1) and NRAC 1–144 (containing id1, tm1, and el) (Fig. 4A). These constructs were co-transfected with CD36 in HEK293T cells, which lack endogenous NRAC and CD36, and immunoprecipitation (IP) of NRAC was performed. The results showed that removal of tm1 abolished the interaction between NRAC and CD36, while deletion of other regions did not impair the interaction (Fig. 4B), indicating that tm1 is essential for NRAC–CD36 binding.

Given that the NRAC–CD36 interaction plays a key role in inhibiting CD36-mediated fatty acid uptake, we next tested whether disrupting this interaction would impair NRAC function. HEK293T cells were co-transfected with CD36 and either full-length NRAC (NRAC 1–165) or the truncated construct lacking tm1 (NRAC 1–112 + SP). While overexpression of full-length NRAC significantly reduced fatty acid uptake, the truncated NRAC construct failed to do so (Fig. 4C). These findings confirm that NRAC's interaction with CD36 is required for its ability to regulate CD36-mediated fatty acid uptake.

Molecular dynamics (MD) simulations are a powerful tool for predicting and analyzing protein–protein interactions at atomic resolution. To explore whether human NRAC and CD36 exhibit similar interaction dynamics as observed in mice, we performed MD simulations of human NRAC (C14ORF180) and CD36. Over the course of 500 ns, the simulations revealed a stable interaction between human NRAC and CD36 (Fig. EV4A–C). Notably, the key interacting residues were predominantly located within the first transmembrane domain (tm1) of human NRAC (Fig. 4D), supporting the experimental findings and highlighting a conserved mechanism of interaction.

## High fat diet feeding resembles NRAC deficiency, redirecting CD36 to clathrin-mediated endocytosis in adipocytes

Based on these data, we hypothesized that high fat diet (HFD) feeding would result in accelerated development of insulin resistance and hyperglycemia in Nrac^ko mice due to preexisting adipocyte hypertrophy and increased fatty acid uptake. Surprisingly, high-fat diet feeding for 10 weeks did not accelerate the development of glucose intolerance (Fig. EV5A,B) systemic insulin resistance (Fig. EV5C), hepatosteatosis (Fig. EV5D–F), or serum markers of adipose inflammation (Fig. EV5G–I) or hepatic function (Fig. EV5J). Conversely, HFD feeding abolished all adipose tissue phenotypes observed in CD fed mice. HFD fed Nrac^ko mice had comparable lipid clearance (Fig. 5A) and serum triglyceride levels (Fig. 5B), with slightly increased HDL levels (Fig. EV5K). HFD fed Nrac^ko mice also showed no differences in body weight (Fig. EV5L), body composition (Fig. 5C,D) and adipocyte size (Fig. EV5M,N). In line with this, fatty acid uptake in primary mature perigonadal (Fig. 5E) and subcutaneous (Fig. 5F) adipocytes from HFD fed mice was not different between wt and Nrac^ko adipocytes.

To understand the reason for the diminished phenotype in HFD fed Nrac^ko mice, we treated mature in vitro differentiated subcutaneous white adipocytes with oleate to mimic the increased fatty acid environment in HFD fed mice. Previous data showed that long-chain fatty acid stimulation leads to CD36 internalization in adipocytes (Hao et al, 2020). Indeed, we observed that acute (Fig. EV6A,B) and chronic (Fig. 5G) fatty acid stimulation leads to increased CD36 internalization in a time and dose-dependent manner.

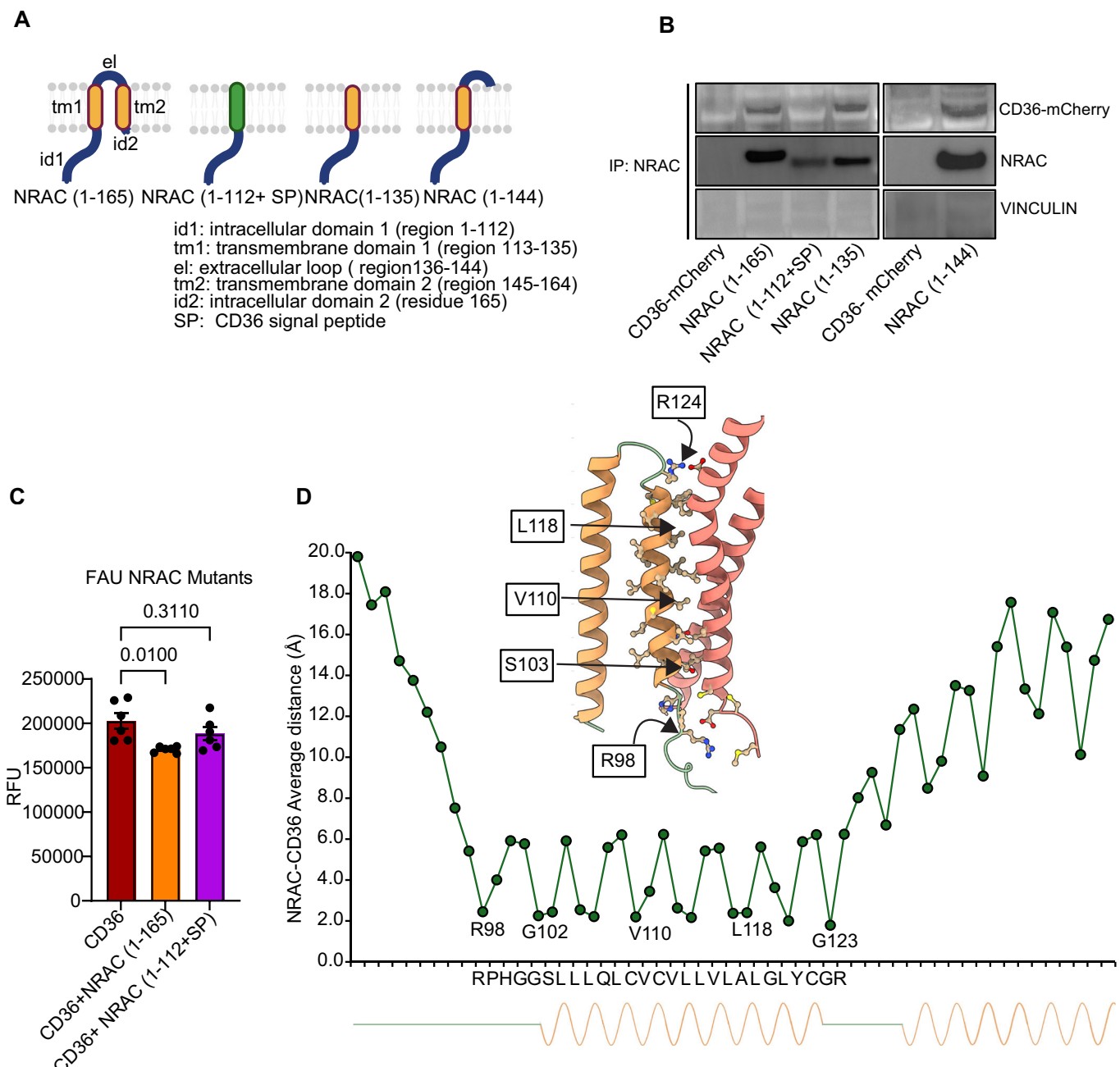

**Figure 4. First transmembrane domain of NRAC is required for its association with CD36.**

(**A**) Cartoon of full-length NRAC and various constructs from mouse NRAC. (**B**) Co-immunoprecipitation of NRAC and CD36. *NRAC* constructs and CD36-mCherry were cotransfected and IP was performed using our NRAC antibody. (**C**) Fatty acid uptake assay in HEK293T cells transfected with either CD36-mCherry alone or cotransfected with NRAC (1–165) and NRAC (1–112 + SP) (*n* = 6). (**D**) The average minimum distance between NRAC residues and CD36 residue was calculated over 500 ns of molecular dynamics simulation. A structural representation of the interaction surface is shown on the top. The first NRAC residue observed to interact with CD36 is R98, located within the intrinsically disordered region (IDR). The first helical interaction begins at G102 and extends to G123, with R124 exhibiting borderline interaction behavior. Data in (**C**) are shown as mean ± SEM. *P < 0.05, **P < 0.01, ***P < 0.001 by one-way ANOVA. Source data are available online for this figure.

In addition, oleate treatment also induced internalization of caveolin-1, in both wt and Nrac^ko adipocytes. Inhibition of clathrin-mediated endocytosis using chlorpromazine restored cell surface localization of CD36 in both wt and Nrac^ko adipocytes but had no effects on caveolin-1 localization (Figs. 5G and EV6C). Importantly, we observed a strongly reduced surface localization of

NRAC upon oleate treatment in wt adipocytes (Figs. 5G and EV6A–C). In line with these in vitro data, immunostaining of NRAC in chow diet and HFD fed mice confirmed that NRAC surface localization is reduced upon HFD feeding in adipose tissue in vivo (Fig. 5H), albeit mRNA expression was increased in both SCF and PGF (Fig. EV6D,E). Female HFD fed mice showed a

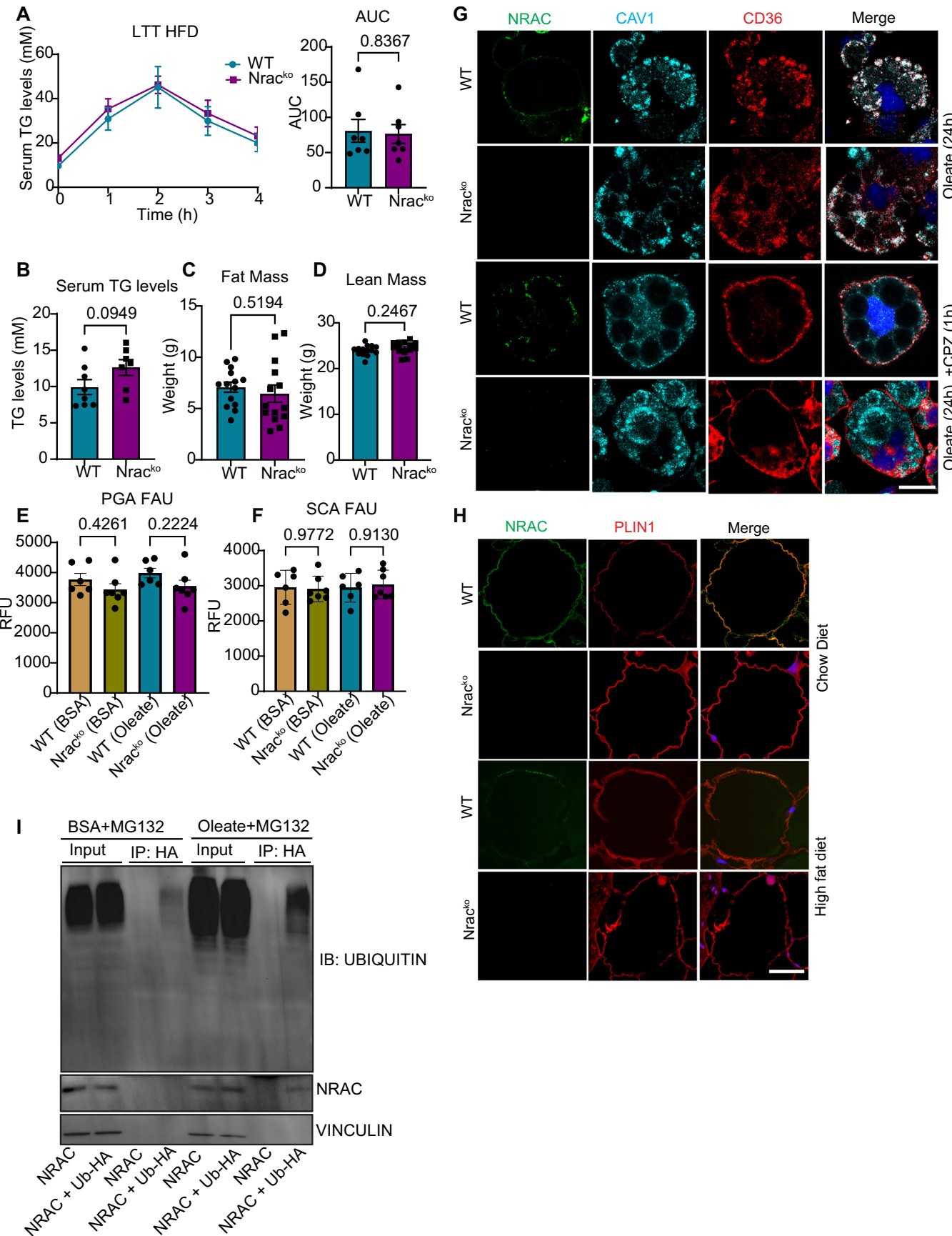

**Figure 5. A high fatty acid environment redirects CD36 to clathrin-mediated endocytosis in adipocytes.**

(A) Lipid tolerance test and area under the curve in mice fed high fat diet ($n = 7$ wt, 7 ko, Age 16 weeks, 10 weeks on HFD), (B) serum triglyceride levels (17 weeks on HFD) ($n = 8$ wt, 7 ko). (C, D) Body composition (13 weeks on HFD). Fatty acid uptake (FAU) assay in primary mature adipocytes from (E) PGF and (F) SCF ($n = 6$ wt, 7 ko). Adipocytes were either treated with BSA or 100 μM oleate. (G) Immunostaining of differentiated adipocytes treated either with oleate (100 μM, 24 h) and ethanol (0.05%, 1 h) or with oleate (100 μM, 24 h) and CPZ (10 μM, 1 h) (scale bar 25 μm). (H) Immunostaining for NRAC and perlipin-1 (PLIN1) on PGF from chow diet (age 23 weeks) and HFD fed male mice (age 23 weeks, 17 weeks on HFD). Scale bar 25 μm. (I) Immunoprecipitation of Ubiquitin-HA. HEK293T cells were transfected with either NRAC or with NRAC + Ubiquitin-HA (Ub-HA) plasmids and treated with with BSA + MG132 (50 μM) or Oleate (100 μM) + MG132 (50 μM) for 1 h. Data are shown as mean ± SEM. *$P < 0.05$, **$P < 0.01$, ***$P < 0.001$ by Student's $t$ test or one or two-way ANOVA. Source data are available online for this figure.

similar expression of NRAC in PGF, However, its expression was significantly reduced following ovariectomy (Fig. EV6F), suggesting that sex hormones influence NRAC expression.

Ubiquitination is a key mechanism regulating the degradation of membrane proteins, and fatty acid treatment is known to induce ubiquitination of various targets (Jeong et al, 2023). To investigate whether the observed increase in *Nrac* mRNA but reduced surface localization is due to NRAC ubiquitination, we transfected HEK293T cells with NRAC in combination with a HA-tagged ubiquitin plasmid. Upon treatment with the proteasome inhibitor MG132, we observed NRAC ubiquitination specifically in the oleate-treated group, but not in the BSA control (Fig. 5I).

These data show that high extracellular fatty acid concentrations result in NRAC ubiquitination and reduced cell surface localization, essentially mimicking NRAC deficiency, promoting a shift from caveolin to clathrin-mediated endocytosis of CD36, promoting fatty acid uptake.

## Human *NRAC* expression and the intronic SNP rs12878589 associate with body fat distribution and obesity

Human *NRAC* (*C14ORF180*) expression data derived from the Leipzig Obesity Biobank (LOBB) comprised of paired samples of omental visceral (VIS) and abdominal subcutaneous (SC) adipose tissues of mostly patients with obesity were analyzed to identify associations of *NRAC* expression and metabolic parameters. We found negative associations between *NRAC* expression in SC adipose tissue and hip circumference, fasting blood glucose, and HbA1c levels (Fig. EV6G). The association with hip circumference appeared to be primarily influenced by female participants, while the negative correlation with HbA1c was significant in male patients but not in females. *NRAC* expression in SC adipose tissue showed a positive correlation with total cholesterol, as well as LDL and HDL cholesterol levels. In general, these associations were reversed in VIS adipose tissue, although they did not reach statistical significance. Conversely, *NRAC* expression in VIS fat was positively associated with body fat percentage, fasting plasma insulin levels, and HOMA-IR. Moreover, we analyzed *C14ORF180* (GRCh38: 104579764–104590515), spanning 10,751 bp with 4370 SNPs, via Biomart and gwasrapidd in R 4.4.1. This identified rs12878589 (14:104584587), showing an association with obesity traits. The intronic variant rs12878589 is nominally associated with chronic laryngitis ($\beta = -0.0921$, $P = 3.23 \times 10^{-3}$) and obesity ($\beta = -0.0315$, $P = 1.99 \times 10^{-2}$) in the FinnGen cohort ($n = 500,349$) (Kurki et al, 2023). Associations were weaker and non-significant in UK Biobank ($n = 420,531$) and MVP African datasets ($n = 121,177$), suggesting possible population-specific effects.

## Discussion

The regulation of adipose lipid uptake and release are essential for metabolic homeostasis and impaired adipocyte lipid handling is a hallmark and underlying cause of the metabolic syndrome (Morigny et al, 2021). Here, we unravel the function of NRAC as rheostat for CD36-mediated fatty acid uptake by controlling the endocytic pathway of CD36 depending on the extracellular fatty acid concentration. In chow diet fed mice, loss of NRAC increases fatty acid uptake resulting in hypertrophic white adipocytes and modestly increased adipose tissue mass. We identified, among others, CD36 and caveolin-1, as NRAC-interacting proteins, indicating that the increased fatty acid uptake is likely the result of altered CD36 cycling. This was subsequently confirmed using in vitro studies of primary white adipocytes showing increased CD36 internalization, independent of caveolin-1, which remained at the cell surface. Moreover, loss of NRAC reduced the interaction of CD36 and caveolin-1, both in vitro and in vivo. Thus, NRAC forms a ternary complex of NRAC/CD36 and caveolin-1 facilitating CD36/ caveolin-1 interaction. Molecular dynamics simulations and studies using NRAC deletion constructs identified an interaction of NRAC and CD36 via the first transmembrane domain of NRAC. In the absence of NRAC the CD36/ caveolin-1 interaction is strongly impaired and CD36 is internalized via clathrin-mediated endocytosis. This results in increased CD36-mediated fatty acid uptake, leading to increased adipocyte size, overall fat mass and adipose tissue weight. While the focus of the current study was on describing the NRAC/CD36 interaction future studies will address a potential direct or indirect interaction of NRAC with caveolin-1. Nevertheless, these changes in Nrac^ko mice did not result in increased adipose tissue inflammation or changes in systemic insulin sensitivity, as seen upon more pronounced adipose hypertrophy (Reilly and Saltiel, 2017). A similar molecular and cell biological phenotype was observed in brown adipocytes. However, the increased fatty acid influx did not result in increased lipid storage but increased mitochondrial respiration and thermogenesis through increased UCP-1 protein levels. Interestingly, BAT showed reduced lipid content in Nrac^ko compared to wt littermate controls. These data suggested an increased BAT activity. However, measurements in metabolic cages did not detect increased energy expenditure at RT, acute cold exposure or acute beta-3-adrenergic receptor activation. The reasons for this are unclear, but the reduced lipid content in BAT could suggest a limited availability of fatty acids for beta oxidation in these chow diet fed animals, which is supported by the reduced serum and BAT triglyceride levels in these mice.

In contrast to this very specific phenotype in chow diet fed animals. HFD feeding abrogated all phenotypic differences, as Nrac^ko mice were not different from HFD fed control animals in all parameters tested. These, at first, negative appearing data, however, allowed us to identify

the main function of NRAC. We show that high fatty acid concentrations result in a dissociation of CD36 from caveolin-1 and internalization in wt adipocytes as observed in Nrac$^{ko}$ adipocytes that can be restored upon inhibition of clathrin-mediated endocytosis. Interestingly, oleate treatment also induced caveolin-1 internalization, which, however, was uncoupled from CD36 and could not be restored using chlorpromazine. Moreover, acute or chronic fatty acid treatment strongly reduced NRAC cell surface localization, which appears to be mediated via ubiquitination and proteasomal degradation of NRAC. Thus, the simple explanation for the abrogation of differences between wt and Nrac$^{ko}$ mice upon HFD feeding is that high extracellular fatty acid concentrations reduce NRAC cell surface localization, essentially mimicking a Nrac$^{ko}$. Our data could also explain previous studies indicating that CD36 internalizes via both clathrin and caveolin-mediated endocytosis, as differences in the extracellular fatty acid concentrations could explain these differences (Hao et al, 2020).

In a broader context, it is important to consider that while the pivotal role of adipose tissue in regulating lipid metabolism is obvious, the precise mechanisms by which adipocytes modulate lipid uptake at the molecular level remain incompletely understood. Regulation of endothelial cell lipoprotein lipase (LPL), cleaving triglyceride rich lipoproteins into free fatty acids (Goldberg et al, 2009; Gonzales and Orlando, 2007) is certainly an important step in regulating adipose tissue lipid uptake. More recent data showed that intracellular palmitoylation of CD36 regulates CD36 cell surface localization and thereby cellular fatty acid uptake (Hao et al, 2020). We now demonstrate that NRAC acts as an adipocyte specific rheostat adjusting CD36 transport kinetics to extracellular fatty acid availability. The adipocyte selective expression of NRAC and the absence of any related genes in the genome, indicate that NRAC has a very specific, yet evolutionary important, function in modulating adipocyte fatty acid uptake. Our analysis of NRAC expression in a human cohort of patients with obesity identified associations that indicate an adipose depot and gender specific role of NRAC in tissue expansion and regulation of systemic insulin sensitivity. However, it is unclear to what extend these mRNA expression data would translate to the modulation of NRAC protein function. It is tempting to speculate that the main function of NRAC is to limit adipocyte fatty acid uptake when circulating levels are low, allowing other tissues, such as the heart that depend on fatty acids, to take up sufficient substrate for beta oxidation. Conversely, when circulating fatty acid levels are high, NRAC gets degraded via ubiquitination, hence, CD36 inhibition is lost, and adipocytes maximize their lipid uptake to prevent toxic ectopic lipid accumulation. It will be interesting to study if pharmacological modulation of NRAC could be utilized to interfere with states if impaired systemic lipid metabolism.

# Methods

### Reagents and tools table

| Reagent/resource | Reference or source | Identifier or catalog number |
|---|---|---|
| **Experimental models** | | |
| C57BL/6N-A530016L24Rik$^{tm1Tigm}$ | Texas A&M genomic medicine | This study |
| 293T | ATCC | CRL-3216 |

| Reagent/resource | Reference or source | Identifier or catalog number |
|---|---|---|
| **Recombinant DNA** | | |
| pCDH-CMV-MCS-EF1α-Puro | Addgene (available from System Biosciences) | Cat# CD510B-1 |
| pCDH-CMV-MCS-EF1α-Nrac (1–165) Puro | This study | |
| pCDH-CMV-MCS-EF1α-Nrac (1–112 + SP) Puro | This study | |
| pCDH-CMV-MCS-EF1α-Nrac (1–135) Puro | This study | |
| pCDH-CMV-MCS-EF1α-Nrac (1–144) Puro | This study | |
| mCherry-CD36-C-10 | Addgene | Plasmid #55011 |
| **Antibodies** | | |
| Mouse Anti-CD36 antibody [JC63.1] | Abcam | Cat# ab23680; RRID:AB_447608 |
| Mouse monoclonal anti-GAPDH | CF-MAB | 1A7 |
| Mouse beta Actin HRP conjugated IgG | Santa Cruz | Cat# sc-47778; RRID:AB_626632 |
| Rabbit polyclonal anti-CD36 | Novus | Cat# NB400; RRID:AB_577623 |
| Rabbit polyclonal anti-Caveolin1 | Cell Signaling | Cat#3238; RRID:AB_2072166 |
| Rabbit polyclonal CD36/SCARB3 Antibody | Sino Biologicals | Cat#80263-T48 |
| Rabbit monoclonal anti-UCP1 IgG | Cell Signaling | Cat#14670; RRID:AB_2687530 |
| Rat monoclonal Nrac antibody | CF-MAB | 11A3 |
| Rat monoclonal anti-GFP IgG | CF-MAB | 3H9 |
| Rat monoclonal Anti-HA IgG | CF-MAB | 3F10 |
| Rabbit monoclonal anti-ubiquitin IgG | Cell signaling | Cat#58395S |
| Vinculin (E1E9V) XP® Rabbit mAb | Cell Signaling | Cat#13901; RRID:AB_2728768 |
| **Oligonucleotides and other sequence-based reagents** | | |
| Oligonucleotides | This study | Table EV1 |
| **Chemicals, enzymes and other reagents** | | |
| Ammonium persulfate (APS) | ThermoFisher | Cat #17874 |
| Antimycin A | Sigma-Aldrich | Cat #A8674 |
| Bovine serum albumin fraction V (BSA) | Carl Roth | Cat #T844.2 |
| Chemiluminescent HRP substrate | Merck Millipore | Cat #WBKLS0500 |
| CL316,243 | Cayman Chemical Company | Cat #17499 |
| Collagenase type IV | Gibco | Cat #17104019 |
| Dako antifade mounting medium | Agilent Technologies | Cat #S302380-2 |

| Reagent/resource | Reference or source | Identifier or catalog number |
|---|---|---|
| DMEM high-glucose + GlutaMAX | Life Technologies | Cat #31966021 |
| Dexamethasone | Sigma-Aldrich | Cat #D4902 |
| Dynabeads™ Protein G | Invitrogen | Cat #10004D |
| EDTA disodium salt dihydrate | Carl Roth | Cat #8043.1 |
| EZ-RUN Recombinant Protein Ladder | Fischer Scientific | Cat #10785674 |
| Fatty acid uptake assay kit | Abcam | Cat# ab287857 |
| Fetal bovine serum | Life Technologies | Cat #10270106 |
| FCCP | R&D Systems | Cat #0453/10 |
| Formaldehyde solution 37% | Carl Roth | Cat #4979.1 |
| D-(+)-Glucose monohydrate | Sigma-Aldrich | Cat #49159 |
| 2- mercaptoethanol | Carl Roth | Cat #4227.3 |
| Hydrochloric acid (HCl 37%) | Sigma-Aldrich | Cat #30721 |
| Intralipid (20%) | Sigma-Aldrich | Cat #I141 |
| Insulin solution human | Sigma-Aldrich | Cat #I9278 |
| Isopropanol | Sigma-Aldrich | Cat #33539 |
| iTaq Universal SYBR® Green Supermix | BioRad | Cat #1725121 |
| 3-Isobuthyl-1-methylxanthin (IBMX) | Sigma-Aldrich | Cat #I5869 |
| Ketamine | Pharmanovo GmbH | Cat #4635 |
| Mayer's solution (Hematoxylin) | Merck | Cat #1092490500 |
| Methanol | Merck Millipore | Cat #UN1230 |
| Normocin | InvivoGen | Cat #Ant-nr-1 |
| Nuclease-free water | Qiagen GmbH | Cat #129114 |
| Nunc™ Lab-Tek™ II Chamber Slide™ System | ThermoScientific | Cat #154534PK |
| NuPAGE™ LDS Sample Buffer (4X) | ThermoFisher | Cat #NP0008 |
| Oligomycin | Sigma-Aldrich | Cat #O4876 |
| Paraffin wax | Leica Surgipath | Cat #39601006 |
| Paraformaldehyde | Carl Roth | Cat #0335.1 |
| Penicillin/Streptomycin (Pen/Strep) | Life Technologies | Cat #15140-122 |
| Phosphatase inhibitor cocktail II | Sigma-Aldrich | Cat #P5726 |
| Phosphatase inhibitor cocktail III | Sigma-Aldrich | Cat #P0044 |
| Phosphate buffer saline (PBS) | Life Technologies | Cat #14190-094 |
| RNeasy Mini Kit | Qiagen | Cat #74106 |
| Rosiglitazone | Santa Cruz Biotechnology | Cat #sc-202795 |
| Rotenone | Sigma-Aldrich | Cat #R8875 |
| Roti®-Histokitt II | Carl Roth | Cat #T160.1 |

| Reagent/resource | Reference or source | Identifier or catalog number |
|---|---|---|
| QIAzol | Qiagen GmbH | Cat #79306 |
| Sodium hydroxide (NaOH) | Carl Roth | Cat #6771.3 |
| Sodium chloride (NaCl) | Carl Roth | Cat #3957.1 |
| Sodium chloride solution 0.9% | Fresenius Kabi Deutschland GmbH | Cat #808765 |
| Sodium dodecyl sulfate (SDS) | Carl Roth | Cat #CN30.3 |
| Sodium hydroxide (NaOH) | Carl Roth | Cat #6771.3 |
| Seahorse XF DMEM Medium | Agilent | Cat #103575-100 |
| StartingBlock™ T20 (TBS) Blocking Buffer | Thermo Fischer | Cat #37543 |
| SuperSignal West FEMTO Max. Sensitivity Substrate | Fisher Scientific | Cat #10187393 |
| Temed | AppliChem | Cat #A1148.0100 |
| Thermo Scientific™ Restore™ PLUS Western Blot Stripping Buffer | ThermoFischer Scientific | Cat #10016433 |
| Triglyceride Assay Kit | Abcam | Cat# ab65336 |
| TRIS base | Carl Roth | Cat #4855.1 |
| Triton™ X-100 | Sigma-Aldrich | Cat #N150 |
| Trypsin, 0.05% EDTA, phenol red | Life Technologies | Cat #11580626 |
| Tween-20 | Santa Cruz Biotechnology | Cat #sc-29113 |
| U-PLEX Custom Metabolic Group | Meso Scale Discovery, Inc | Cat #K151ACM-2 |
| Xylene | Carl Roth | Cat #9713.5 |
| **Softwares** | | |
| ImageJ | https://imagej.net | RRID: SCR_003070 |
| GraphPad Prism 10 | https://www.graphpad.com/ | RRID: SCR_002798) |
| Adobe Illustrator | https://www.adobe.com/products/illustrator.html | RRID: SCR_010279 |

## Animals

Nrac knockout mice were obtained from Texas A&M genomic medicine. Mice were generated by replacing the first and second coding exons with an IRES/bGeo/PolyA cassette via homologous recombination. Heterozygous mice (Nrac$^{+/-}$) were mated to obtain WT (Nrac$^{+/+}$) or Nrac$^{ko}$ (Nrac$^{-/-}$) mice. Littermates were used for experiments. Genotyping was performed as suggested by Texas A&M genomic medicine. Mice were bred on a C57BL/6J background and maintained in a conventional animal facility at a constant ambient temperature of $22 \pm 2\,°C$, with 45–65% humidity and a 12-h light–dark cycle in ventilated racks. The mice were fed either a standard chow diet (Altromin 1314, Lage Germany) or a 58% high-fat diet (HFD) (Research Diets D12331, New Brunswick (NJ), USA). Body weights were determined at

weekly intervals and body composition was measured by nuclear magnetic resonance measurements (EchoMRI ®LLC, Houston, USA) before the onset of the HFD feeding and prior to sacrificing the mice. Animal experiments were conducted in accordance with the German animal welfare law and performed with permission and in accordance with all relevant guidelines and regulations of the district government of Upper Bavaria (Bavaria, Germany) (Animal protocol numbers ROB-55.2-2532.Vet-2532.Vet_02-18-188, ROB-55.2-2532.Vet-2532.Vet_02-17-232 ROB-55.2-2532.Vet_02-20-202, ROB-55.2Vet-2532.Vet_03-17-68 and ROB-55.2-2532.Vet_02-14-33).

### Ovariectomy

For bilateral ovariectomy (OVR) or Sham-operation, mice were anesthesized (Ketamine: 100 mg/kg, Xylazine: 7 mg/kg, intraperitoneal) and received analgesia (200 mg/kg Metamizol, subcutaneously), both flanks were shaved, and mice were placed on their right side. Following, an incision was made through first skin and then the muscle layers in the left flank caudal to the last rib. The ovary was then carefully pulled out and removed after clamping between the distal part of uterine horn and the ovary. After returning the horn into the abdominal cavity, the wound was closed by suturing first the muscle layer and then the skin with surgical silk. The same procedure was applied to remove the right ovary. Sham surgeries were also performed, in which each ovary was exposed but not dissected. For postoperative pain management 5 mg/kg meloxicam was injected subcutaneously right after the surgery and twice daily in the following 2 days. HFD exposure was started one week after the surgery.

## Metabolic phenotyping

For the glucose tolerance test (GTT) or insulin tolerance test (ITT), mice were fasted for 4 h and 2 g/kg glucose or 0.75 IU/kg insulin, respectively, were injected intraperitoneally. Blood glucose values were checked after tail puncture using a handheld glucometer (Abott, #47081899). Lipid tolerance tests (LTT) were performed as previously described (Peterson et al, 2014). Mice were fasted for 16 h and 20%-emulsified lipid (intralipid) was injected intraperitoneally, and serum triglycerides levels were measured at 1 h interval for 4 h using triglyceride assay kit (Abcam). For lipoprotein separation, samples were pooled and analyzed via fast-performance liquid chromatography (FPLC) gel filtration as described previously (Hofmann et al, 2008). Cholesterol levels in FPLC fractions were measured using the Amplex red cholesterol assay kit (Invitrogen).

## Isolation of adipocytes and the stromal vascular fraction (SVF), preadipocyte immortalization and fatty acid uptake

### Preadipocyte immortalization

Freshly isolated adipose tissues from WT and Nrac[ko] littermates were chopped using a surgical knife and a scissor and digested in the digestion medium containing DMEM, 1% BSA and 1 mg/ml Collagenase IV for 30 min at 1000 rotations per minute in a thermo-shaker at 37 °C. The cells were then filtered through a 100-μm filter and centrifuged at $300 \times g$ for 5 min, washed once with the cell culture medium and seeded on six-well plates. Cells were immortalized using an ecotropic SV40 large T retrovirus on the day after isolation.

### Generation of stable cell line

NRAC-pCDH-CMV-MCS-EF1α and pCDH-CMV-MCS-EF1α-Puro (scrambled control) were packaged into lentiviruses using the ViraSafe™ Lentiviral Packaging System (Cell Biolabs, VPK-205) and concentrated with PEG-it Virus Precipitation Solution (System Biosciences, LV810A-1). Immortalized subcutaneous preadipocytes from wt and Nrac[ko] mice were infected in the presence of 9 μg/mL polybrene. Following infection, cells were cultured in Dulbecco's Modified Eagle Medium (DMEM; Thermo Fisher Scientific, 35050038) supplemented with 10% fetal bovine serum, 1% penicillin–streptomycin (Thermo Fisher Scientific, 5000956), and 2.5 μg/mL puromycin (Cayman Chemical, Cay13884-500).

### Vector construction and transfection

The pCDH_CMV_MCS_EF1_Puro vector (Addgene, CD510B-1) was digested with EcoRI and NheI-HF (NEB) in the presence of rCutSmart™ Buffer, followed by dephosphorylation using rSAP. Digested products were resolved on a 1% agarose gel, and the linearized vector was purified using the QIAquick Gel Extraction Kit (Qiagen). gBloc fragments were similarly digested and purified using the QIAquick PCR Purification Kit. Ligation was performed using T4 DNA Ligase (NEB) at a 1:12 vector-to-insert molar ratio and incubated at room temperature for 10 min. Constructs were verified by restriction digestion with PvuI-HF and confirmed by Sanger sequencing (Eurofins Genomics, Germany). HEK293T cells were transfected with 2 μg of plasmid DNA using PolyFect Transfection Reagent (Qiagen) according to the manufacturer's protocol and incubated for 24 h.

### Isolation and differentiation of primary adipocytes and SVF

Mature adipocytes were isolated and cultured as described in (Villanueva-Carmona et al, 2023). Briefly, after digestion with digestion medium, described above, the cell suspension was filtered through a 250-μm filter and the cells were let stand for 20 min. The upper layer containing mature adipocytes was collected and washed once with PBS. After the wash, the mature adipocytes were seeded in 96-well plates and used for the fatty acid uptake assay (Abcam, ab287857). For differentiating primary preadipocytes, the SVF was washed once with cell culture medium, and the cells were counted and seeded either on immunostaining chamber slides or seahorse plates. Preadipocytes were differentiated using induction medium cocktail of IBMX (0.5 mM), insulin (100 nM), Dexamethasome (5 μM), indomethacin (125 μM) in DMEM 10% FBS for 2 days and afterwards the medium was changed every 2 days to differentiation medium (DMEM, 10% FBS) containing 100 nM insulin. The differentiated adipocytes were trypsinized and plated onto poly-D-lysine coated plates and after serum starvation for 1 h, the cells were treated with either DMSO or with 5 μM isoproterenol for 20 mins and the kit protocol was followed to measure the fluorescence intensity (Abcam, ab287857).

### Oleate and chlorpromazine treatment of primary and immortalized in vitro differentiated adipocytes

Twenty millimolar sodium oleate was dissolved in 150 mM NaCl at 70 °C. To conjugate with fatty acid-free BSA, the sodium oleate was

added dropwise in 10% BSA (prepared in 150 mM NaCl) at 37 °C to get a final 1:1 Sodium oleate and BSA mixture (final concentration of Sodium oleate 10 mM in 5% BSA) and pH was set at 7.4. Chlorpromazine was dissolved in ethanol to prepare a stock solution of 20 mM and further diluted in DMEM. On day 8 of differentiation, cells were starved for four hours in serum-free DMEM and 100 nM insulin and treated with either 100 μM oleate or chlorpromazine as described in the figure legends.

## Generation of a monoclonal antibody against mouse NRAC

A peptide representing aa 29-42 of NRAC was synthesized and coupled to ovalbumin or biotin (Peps4LS, Heidelberg, Germany). Lou/c rats were immunized subcutaneously (s.c.) and intraperitoneally (i.p.) with a mixture of 40 μg OVA-coupled peptides in 400 μl PBS, 5 nmol CpG2006 (TIB MOLBIOL, Berlin, Germany), and 400 μl incomplete Freund's adjuvant. After 6 weeks, a boost without Freund's adjuvant was given i.p. and s.c. 3 days before fusion. Fusion of the myeloma cell line P3X63-Ag8.653 with the immune spleen cells was performed according to standard procedure (Kohler and Milstein, 1975). After fusion, the cells were plated in 96-well plates using RPMI 1640 with 20%fetal calf serum, glutamine, pyruvate, non-essential amino acids and HAT media supplement (Hybri-Max, Sigma-Aldrich). Hybridoma supernatants were screened 10 day later by ELISA on biotinylated peptides and binding was detected using HRP-labelled isotype-specific monoclonal mouse-anti-rat IgG secondary antibodies. Positive supernatants were further validated by Western blot analysis and hybridoma cells of clone NRAC 11A3 (rat IgG2a) were subcloned by limiting dilution to obtain a stable monoclonal cell line.

## RNA Isolation from whole tissue

Adipose tissues were lysed with Qiazol (Qiagen) using a TissueLyzer II (Qiagen). After incubating for 5 min at RT, chloroform was added to the samples and vortexed for 15 sec and let sit at RT for 3 min. Homogenates were then centrifuged at $12,000 \times g$ for 15 min at 4 °C. The clear aqueous phase was transferred to a new Eppendorf tube and mixed with same volume of 70% ethanol. RNA isolation was performed according to manufacturer's protocol using RNeasy Mini Kit (Qiagen). Total RNA concentrations were measured at 260 nm using NanoDrop 2000 UV–Vis Spectrophotometer (Thermo Scientific).

## cDNA synthesis and RT-qPCR Analysis

Five hundred nanograms of total RNA was reverse transcribed to cDNA using the High-Capacity cDNA Reverse Transcription Kit (Applied Biosystems) following the manufacturers' protocol. Relative mRNA expression was quantified by Real-Time semi quantitative polymerase chain reaction (RT-qPCR). cDNA, iTaq Universal SYBR® Green Supermix (BioRad) and 300 nM forward and reverse primers (see below) were mixed and run on a C1000 Touch Thermal Cycler (BioRad) in duplicates or triplicates and quantified using Bio-Rad CFX Manager 3.1 Program. All expressions levels were normalized to TATA-binding protein (*Tbp*). Primers sequences are listed in Table EV1.

## Western blotting

Cells were lysed in ice-cold modified RIPA buffer [50 mM Tris (pH 7.4), 150 mM NaCl, 1 mM EDTA, 1% Triton X-100] containing protease and phosphatase inhibitors and incubated with 0.1% SDS on ice for 10 min and centrifuged to remove debris. Protein concentration was measured using a BCA assay (Thermo Fisher Scientific, 10741395). Protein lysates were then mixed with 4×LDS Sample Buffer (NuPAGE™ Thermo Fisher Scientific, NP0008) with 2.5% 2-mercaptoethanol and incubated at 70 °C for 10 min. The protein samples were separated via SDS-PAGE using 4-20% gradient gels in 1×Tris-Glycine running buffer containing 10% SDS.

Proteins were transferred to 0.2-μm PVDF membranes (Merck Millipore, ISEQ00010), and blocked with either 5% BSA (Carl Roth, T844.2) or with 5% skim milk (Biomol, S1013-90A.500) in 10 mM Tris TBS-0.1% Tween 20 (TBST) for 1 h at room temperature. Primary antibodies were diluted with Starting blocking buffer and HRP-conjugated secondary antibodies were diluted in 5% skim milk in TBST. Amersham Hyperfilm ECL (GE, GE28-9068-36) and HRP substrate ECL (Merck Millipore, WBKLS0500) were used to detect signals using a Biorad ChemiDoc imaging system. Band intensities were quantified by ImageJ.

## Immunoprecipitation and mass spectrometry

Cells or tissues were lysed in ice-cold IP lysis buffer (25 mM Tris HCl, 1 mM EDTA, 150 mM NaCl, 5% Glycerol and 1% Triton X-100) containing protease and phosphatase inhibitors. Four milligram of total protein lysate was incubated with the antibody and incubated overnight on rotation at 4 °C. The lysates with the antibody were then incubated with Dynabeads™ (Invitrogen) for 1 h at 4 °C on rotation. The beads were washed three times using the IP lysis buffer and eluted with 1× LDS buffer containing 2.5% 2-mercaptoethanol at 70 °C for 10 min for immunoblotting or the beads were washed with IP lysis buffer (without glycerol) and eluted with Laemmli buffer for further mass spectrometric analysis or western blotting.

## Sample preparation for mass spectrometry

Proteins were subjected to tryptic digestion using S-Trap mini spin columns (Protifi, USA) with a modified preparation procedure. Initially, proteins were extracted in 50 μL of Laemmli buffer. After protein reduction and alkylation employing DTT and iodoacetamide, phosphoric acid was added to achieve a final concentration of 1.2%. Subsequently, this mixture was combined with six volumes of binding buffer (90% MeOH containing a final concentration of 100 mM Tris at pH 7.1). Following mixing, the protein solution was loaded onto the filter and centrifuged at $1000 \times g$ for 1 min. The samples were then washed three times with the binding buffer. For the digestion step, 40 μL of 50 mM ammonium bicarbonate containing 0.5 μg of Lys-C (Wako) were added to the filter, and the mixture was incubated for 2 h at room temperature, followed by an additional incubation for 16 h at 37 °C using 1 μg of trypsin (Promega). Peptides were collected by centrifugation and acidified with 0.5% trifluoroacetic acid (TFA) and used for LC-MSMS.

## LC-MS/MS analysis

Liquid chromatography with tandem mass spectrometry (LC-MSMS) analysis was performed in data-dependent acquisition (DDA) mode. MS data were acquired on a Q-Exactive HF-X mass spectrometer (Thermo Fisher Scientific, Waltham, Massachusetts, USA) each online coupled to a nano-RSLC (Ultimate 3000 RSLC; Dionex). Tryptic peptides were automatically loaded on a C18 trap column (300 µm inner diameter (ID) × 5 mm, Acclaim PepMap100 C18, 5 µm, 100 Å, LC Packings) at 30 µl/min flow rate. For chromatography, a C18 reversed-phase analytical column (nanoEase MZ HSS T3 Column, 100 Å, 1.8 µm, 75 µm × 250 mm, Waters) was used at a flow rate of 250 nl/min and following a 95-min non-linear acetonitrile gradient from 3 to 40% in 0.1% formic acid. The high-resolution (60,000 full width at half-maximum) MS spectrum was acquired with a mass range from 300 to 1500 $m/z$ with an automatic gain control target set to $3 \times 10^6$ and a maximum of 30 ms injection time. From the MS prescan, the 15 most abundant peptide ions were selected for fragmentation (MS-MS) if at least doubly charged, with a dynamic exclusion of 30 s. MS-MS spectra were recorded at 15,000 resolutions with an automatic gain control target set to $5 \times 10^2$ and a maximum of 50 ms injection time. The normalized collision energy was 28, and the spectra were recorded in profile mode.

### Data processing-protein identification

Proteome Discoverer 2.5 software (Thermo Fisher Scientific; Waltham, Massachusetts, USA, version 2.5.0.400) was used for peptide and protein identification via a database search (Sequest HT search engine) against Swissprot MOUSE database (Release 2020_02, 17061 sequences). Search settings were 10 ppm precursor tolerance, 0.02 Da fragment tolerance, one missed cleavage allowed. Carbamidomethylation of Cys was set as a static modification. Dynamic modifications included deamidation of Asn, Gln and Arg, oxidation of Pro and Met; and a combination of Met loss with acetylation on protein N-terminus. Percolator was used for validating peptide spectrum matches and peptides, accepting only the top-scoring hit for each spectrum, and satisfying the cutoff values for false discovery rate (FDR) < 1%, and posterior error probability < 0.01. Protein quantification was determined by utilizing abundance values for unique peptides and normalization was based on the total peptide amount. For the quantification process, the three most abundant peptides (3 N) were considered, and the ratio calculation was based on protein abundance, with a maximum fold change threshold of 100. For statistical testing an ANOVA was used.

## Immunostaining

The preadipocytes were seeded on eight-well chamber slides (Thermofischer) and differentiated for 8 days. On day 8, after the respective treatments, cells were washed with PBS and fixed in 4% PFA (pH 7.4) for 10 min at room temperature and blocked and permeabilized using 3% BSA and 0.3% Tween 20. Cells were then incubated with anti-CD36 (1:250, Abcam, ab23680), anti-CAV1 (1:250, CST, 3238 s), anti-NRAC (1:10, 11A3) overnight at 4 °C followed by three times wash with PBS, and then incubated with Alexa Fluor 488 goat anti-rabbit IgG (1:500, Thermo Fisher Scientific), Alexa Fluor 594 goat anti-mouse IgG (1:500, Thermo

Fisher Scientific) and Alexa Fluor 647 goat anti-rat IgG (1:500). After washing three times, slides were mounted using Dako antifade mounting medium and imaged on a Leica TCS SP8 microscope.

## Measurement of cellular oxygen consumption rate (OCR)

Primary brown preadipocytes were isolated as described above and differentiated using induction medium (DMEM 10% FBS) containing IBMX (0.5 mM), insulin (100 nM), Dexamethasome (5 µM), indomethacin (125 µM), T3 (1 nM) and Rosiglitazone (1 µM) for 2 days and afterwards the medium was changed every 2 days to differentiation medium (DMEM 10% FBS) containing 100 nM insulin, T3 and rosiglitazone until day 8. On day 8 of differentiation, OCR was measured with a XF96 Extracellular Flux analyzer (Seahorse Bioscience). Cells were equilibrated in assay medium [Agilent Seahorse XF DMEM supplemented with 0.2% fatty acid-free BSA, 25 mM glucose (Sigma-Aldrich), 2 mM GlutaMAX (Gibco)] for 1 h prior to measurement at 37 °C in a $CO_2$ free incubator. Compounds were prepared at 10-times concentration in respiration base media (assay media without FFA-free BSA) and loaded to the overnight equilibrated cartridge ports (A: 50 µM Isoproterenol, B:50 µM Oligomycin 50 µM, C: 75 µM FCCP, D: 50 µM Antimycin A and 30 µM Rotenone). The cartridge was equilibrated for at least 20 min in the XF96 Extracellular Flux analyzer. Each cycle was consisting of 4 min of mixing, and 2 min of measuring. Measurement was recorded before and after each injection for three cycles. Each cycle per biological replicate was averaged and plot on a graph against time. The immortalized brown preadipocytes were also differentiated the same way and the injection strategies has been mentioned in the respective figure legends.

## Biomarkers MSD

A multiplex assay platform was used to measure the concentrations of IL-6, TNFα, Leptin and FGF-21 in plasma samples by electrochemiluminescence-linked immunosorbent assay (ECLIA) based on Mesoscale Discovery (MSD) technology (U-PLEX Custom Metabolic Group). A 10-spot MSD plate is coated with specific antibodies that have been pre-treated with corresponding spot-linkers. Plasma samples are diluted 1:2 and incubated for 1 h. Following this, the samples are incubated for another hour with conjugated detection antibodies before being analyzed using the MSD plate reader. The antibodies are provided as U-Plex antibodies from MSD. Data analysis is performed using MSD Discovery Workbench software.

## Histology

Immediately after dissection, adipose tissues and liver were fixed in 4% paraformaldehyde (PFA, Carl Roth, Germany) 4 °C for 24 h and then dehydrated in an ascending concentration of ethanol (70–100%) and xylene. Afterwards, tissues were embedded with paraffin (Leica, Germany) and cut into 2-µm sections using a microtome (Leica, Germany). Dissected tissue sections were stained with hematoxylin and eosin (H&E). The images were taken with a Nikon microscope and the adipocyte sizes were quantified using Image J plugin Adiposoft.

## Human data

The human data utilized in this study were obtained from the cross-sectional cohort (CSC) of the Leipzig Obesity Biobank (LOBB, https://www.helmholtz-munich.de/en/hi-mag/cohort/leipzig-obesity-bio-bank-lobb). This cohort includes paired samples of abdominal subcutaneous and omental visceral adipose tissue from a total of 1,479 participants. These individuals are characterized as either having normal/overweight ($N = 31$; 52% women; average age: $55.8 \pm 13.4$ years; average BMI: $25.7 \pm 2.7$ kg/m²) and obese ($N = 1448$; 71% women; average age: $46.9 \pm 11.7$ years; average BMI: $49.2 \pm 8.3$ kg/m²). Adipose tissue samples were collected during elective laparoscopic abdominal surgeries, adhering to established protocols (Langhardt et al, 2018; Mardinoglu et al, 2015). Body composition and metabolic parameters were evaluated using standardized methods as previously detailed (Bluher, 2020; Kloting et al, 2010). Exclusion criteria for participation included being under 18 years of age, having a history of chronic substance or alcohol abuse, smoking within the last 12 months prior to surgery, suffering from acute inflammatory diseases, using glitazones concurrently, having end-stage malignancies, experiencing weight loss greater than 3% in the three months before surgery, uncontrolled thyroid disorders, and Cushing's disease. The study received approval from the Ethics Committee of the University of Leipzig (approval number: 159-12-21052012) and was conducted in accordance with the principles set forth in the Declaration of Helsinki. All participants provided written informed consent prior to their inclusion in the study.

We generated ribosomal RNA-depleted RNA sequencing data using the SMARTseq protocol (Picelli et al, 2014; Song et al, 2018). The libraries were sequenced as single-end reads on a Novaseq 6000 (Illumina, San Diego, CA, USA) at the Functional Genomics Center in Zurich, Switzerland. The preprocessing steps were performed as previously outlined (Hagemann et al, 2023). In summary, adapter and quality-trimmed reads were aligned to the human reference genome (assembly GRCh38.p13, GENCODE release 32), and gene-level expression quantification was conducted using Kallisto (Bray et al, 2016) version 0.48. For samples with read counts exceeding 20 million, we downsampled them to 20 million reads using the R package ezRun version 3.14.1 (https://github.com/uzh/ezRun, accessed on April 27, 2023). Data normalization was performed using a weighted trimmed mean (TMM) of the log expression ratios, incorporating adjustments for age, transcript integrity numbers (TINs), and gender, except in cases where male and female samples were analyzed independently. All analyses were performed in R version 4.3.1 (www.R-project.org).

## MD simulations

### Initial structure generation
NRAC lacks a resolved crystal structure, but several UniProt entries exist for different species (Consortium, 2024). CD36, in complex with the CIDRα domain from MCvar1 PfEMP1 (PDB-ID: 5LGD), has been crystallized. However, the specific region of interest—where NRAC is likely to bind is unresolved so needed computational modeling (Hsieh et al, 2016). The human sequences of each protein were used inAlphaFold2 predictions (Jumper et al, 2021).

Experimental evidence suggests that binding occurs within the membrane, involving residues in NRAC's TM1 region and a transmembrane helix of CD36. The transmembrane topology was predicted using DeepTMHMM (Hallgren et al, 2022), and the membrane orientation was determined with the PPM server (Lomize et al, 2012). Two potential interaction sites were identified: NRAC residues 105–121 and CD36 residues 8–30 or 440–466.

Full-length 3D structures of both proteins were generated using the LIT-AlphaFold pipeline with AlphaFold2 (Mirdita et al, 2022) (Urvas et al, 2024) (Yu et al, 2023). From a visual inspection of PPM's output, the most promising interacting residues were selected, i.e., residues 105–121 for NRAC and 14–30 or 440–456 for CD36. These regions were docked using HADDOCK2.4 (Honorato et al, 2024), and the top clusters were selected for molecular dynamics simulations.

### Molecular dynamics simulations
Molecular dynamics simulations were conducted to assess the stability of the protein–protein complexes. Simulations were performed using GROMACS (Abraham et al, 2015) with the CHARMM36 force field (Huang and MacKerell Jr, 2013). As an initial step, explicit water simulations were run to evaluate complex stability in aqueous solution, given the largely lipophilic nature of the interaction surface. System preparation and execution followed standard protocols (Lemkul, 2018).

Subsequently, the complexes were embedded in a lipid bilayer using CHARMM-GUI (Lee et al, 2016) for all-atom membrane simulations. Multiple membrane compositions were tested; the final system comprised 40 cholesterol and 110 POPC (1-palmitoyl-2-oleoyl-sn-glycero-3-phosphocholine) molecules per leaflet. All simulations were run for 500 ns. Analysis was performed using custom Python scripts, and visualizations were generated with ChimeraX (Meng et al, 2023).

## Statistical analysis

All statistics were calculated using GraphPad Prism 10. Data are presented as mean ± standard error of the mean (SEM) unless stated differently in the figure legend. Statistical significance was determined by unpaired Student's $t$ test or, using one- or two-way analysis of variance (ANOVA), followed by Tukey's multiple comparison test if not otherwise stated in figure legends. Differences reached statistical significance with $P < 0.05$. The Spearman correlation coefficient was employed to evaluate the relationship between human *C14ORF180* (human *NRAC*) expression and metabolic parameters, with adjustments made for multiple testing using the false discovery rate.

## Data availability

The data supporting the findings of this study are available within the paper and its supplementary information files. The human RNA-seq data from the LOBB have not been deposited in a public repository due to restrictions imposed by patient consent but can be obtained from Matthias Blüher upon request. LIT-AlphaFold can be found at https://github.com/LIT-CCM-lab/LIT-AlphaFold. The code of AlphaFold, as well as the related models' weights, genetic databases, and template database can be found at https://github.com/google-deepmind/alphafold. The mass spectrometry data are deposited to the ProteomeXchange Consortium (http://

proteomecentral.proteomexchange.org) via the PRIDE partner repository with the dataset identifier PXD PXD054842.

The source data of this paper are collected in the following database record: biostudies:S-SCDT-10_1038-S44318-025-00520-2.

## Peer review information

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

## Acknowledgements

We thank Vignesh Karthikaisamy and Mauricio Berriel Diaz for providing the pRK5-HA-Ubiquitin-WT plasmid. IS was supported by the National Overseas Scholarship from the Government of India. YO received support through the Post-doctoral Fellowship program from The Uehara Memorial Foundation, Japan (201830047) and the Alexander von Humboldt-Stiftung, Germany. LK, CW and XY were supported by China Scholarship Council. DK and IT got funding from the Horizon Europe funding programme under the Marie Skłodowska-Curie Actions Doctoral Networks grant agreement "Explainable AI for Molecules - AiChemist", no. 101120466. German Mouse Clinic received grant from German Federal Ministry of Education and Research (Infrafrontier grant 01KX1012 to MHdA) and German Center for Diabetes Research (DZD) (MHdA). MB received funding from grants from the DFG Projektnummer 209933838—SFB 1052 (project B1) and by Deutsches Zentrum für Diabetesforschung (DZD, Grant: 82DZD00601). We thank Wenfei Sun and Hua Dong for supporting human adipose tissue RNA sequencing.

## Author contributions

**Inderjeet Singh**: Conceptualization; Data curation; Formal analysis; Validation; Investigation; Visualization; Methodology; Writing—original draft; Project administration; Writing—review and editing. **Yasuhiro Onogi**: Conceptualization; Supervision; Investigation; Writing—review and editing. **Filipe Menezes**: Formal analysis; Investigation; Methodology; Writing—review and editing. **Dina Khasanova**: Investigation; Methodology; Writing—review and editing. **Lingru Kang**: Investigation; Methodology; Writing—review and editing. **Chenxi Wang**: Investigation; Methodology; Writing—review and editing. **Julio Ruiz-Trave**: Formal analysis; Investigation; Methodology; Writing—review and editing. **Sapna Sharma**: Formal analysis; Investigation; Methodology; Writing—original draft; Writing—review and editing. **Ahmed Khalil**: Investigation; Methodology; Writing—review and editing. **Valentin K Reichenbach**: Formal analysis; Investigation; Visualization; Methodology; Writing—review and editing. **Yingzi Shi**: Investigation; Methodology; Writing—review and editing. **Andrew Flatley**: Formal analysis; Validation; Investigation; Methodology; Writing—review and editing. **Xiaocheng Yan**: Investigation; Methodology; Writing—review and editing. **Andreas Israel**: Investigation; Methodology; Project administration; Writing—review and editing. **Nathalia RV Dragano**: Data curation; Formal analysis; Investigation; Methodology; Writing—review and editing. **Juan Antonio Aguilar-Pimentel**: Conceptualization; Data curation; Formal analysis; Investigation; Methodology; Writing—review and editing. **Anne Hoffmann**: Data curation; Formal analysis; Investigation; Methodology; Writing—review and editing. **Adhideb Ghosh**: Formal analysis; Investigation; Methodology; Writing—review and editing. **Falko Noé**: Formal analysis; Investigation; Methodology; Writing—review and editing. **Christian Wolfrum**: Supervision; Methodology; Writing—review and editing. **Sebastian Cucuruz**: Formal analysis; Investigation; Methodology; Writing—review and editing. **Ann-Christine König**: Formal analysis; Investigation; Methodology; Writing—review and editing. **Ingo Burtscher**: Investigation; Methodology; Writing—review and editing. **Stefanie M Hauck**: Supervision; Writing—review and editing. **Heiko Lickert**: Supervision; Writing—review and editing. **Susanna M Hofmann**: Supervision; Methodology; Writing—review and editing. **Regina Feederle**: Supervision; Methodology; Writing—review and editing. **Sonja C Schriever**: Investigation; Writing—review and editing. **Rene Hernandez-Bautista**: Investigation; Methodology; Writing—review and editing. **Gencer Sancar**: Funding acquisition; Investigation; Methodology; Writing—review and editing. **Alberto Cebrian-Serrano**: Supervision; Investigation; Methodology; Writing—review and editing. **Igor Tetko**: Supervision; Writing—review and editing. **Helmut Fuchs**: Supervision; Methodology; Writing—review and editing. **Valérie Gailus-Durner**: Supervision; Methodology; Writing—review and editing. **Matthias Blüher**: Data curation; Formal analysis; Supervision; Methodology; Writing—review and editing. **Martin Hrabé de Angelis**: Supervision; Writing—review and editing. **Siegfried Ussar**: Conceptualization; Formal analysis; Supervision; Funding acquisition; Validation; Visualization; Writing—original draft; Project administration; Writing—review and editing.

Source data underlying figure panels in this paper may have individual authorship assigned. Where available, figure panel/source data authorship is

listed in the following database record: biostudies:S-SCDT-10_1038-S44318-025-00520-2.

## Funding

## Disclosure and competing interests statement

MB received honoraria as a consultant and speaker from Amgen, AstraZeneca, Bayer, Boehringer-Ingelheim, Lilly, Novo Nordisk, Novartis, and Sanofi. All other authors declare no conflict of interest. The funders had no role in the design of the study; in the collection, analyses, or interpretation of data; in the writing of the manuscript; or in the decision to publish the results.

# Expanded View Figures

**Figure EV1.  NRAC deletion does not affect glucose metabolism or insulin sensitivity in chow diet fed mice.**

Validation of NRAC antibody by (**A**) western blotting of HEK293T cells transfected with either NRAC-GFP or GFP plasmids. Untransfected cells were used as control. (**B**) Western blotting to detect endogenous NRAC in BAT, SCF and PGF and (**C**) immunostaining of subcutaneous differentiated mature adipocytes to detect endogenous NRAC (scale bar 10 µm), and (**D**) Immunoprecipitation of NRAC from perigonadal fat (PGF). (**E–H**) mRNA expression of *Tnfa* and *Il6* in PGF and SCF of male mice ($n = 6$ wt, 5ko), (**I**) weights of liver, heart and pancreas ($n = 9$ wt, 5ko). (**J**) Glucose tolerance test and area under the curve (age 8 weeks). (**K**) Insulin tolerance test (ITT) and area under the curve (AUC) in 9 weeks old male mice. (**L**) Percentage of HbA1c in 12-week-old male mice. (**M**) Body weight development of chow diet fed female wt and Nrac$^{ko}$ mice. (**N**) Fat and (**O**) lean mass of 15-week-old chow diet fed female mice. (**P**) Glucose tolerance test and area under the curve of 8 weeks old female mice. (**Q**) %HbA1c in 12-week-old female mice. (**R**) Insulin tolerance test (ITT) and area under the curve (AUC) in 9-week-old female mice. (**S**) H&E staining of SCF and PGF tissues of female mice (Age 23 weeks), and (**T**) quantification of lipid droplets. $n = 9$wt and 7ko for males and 5wt and 7ko for females. (**U**) mRNA expression of *Nrac* from BAT, SCF and PGF of male and female mice ($n = 4$). Data are shown as mean ± SEM. *$P < 0.05$, **$P < 0.01$, ***$P < 0.001$ by Student's *t* test or one or two-way ANOVA. Source data are available online for this figure.

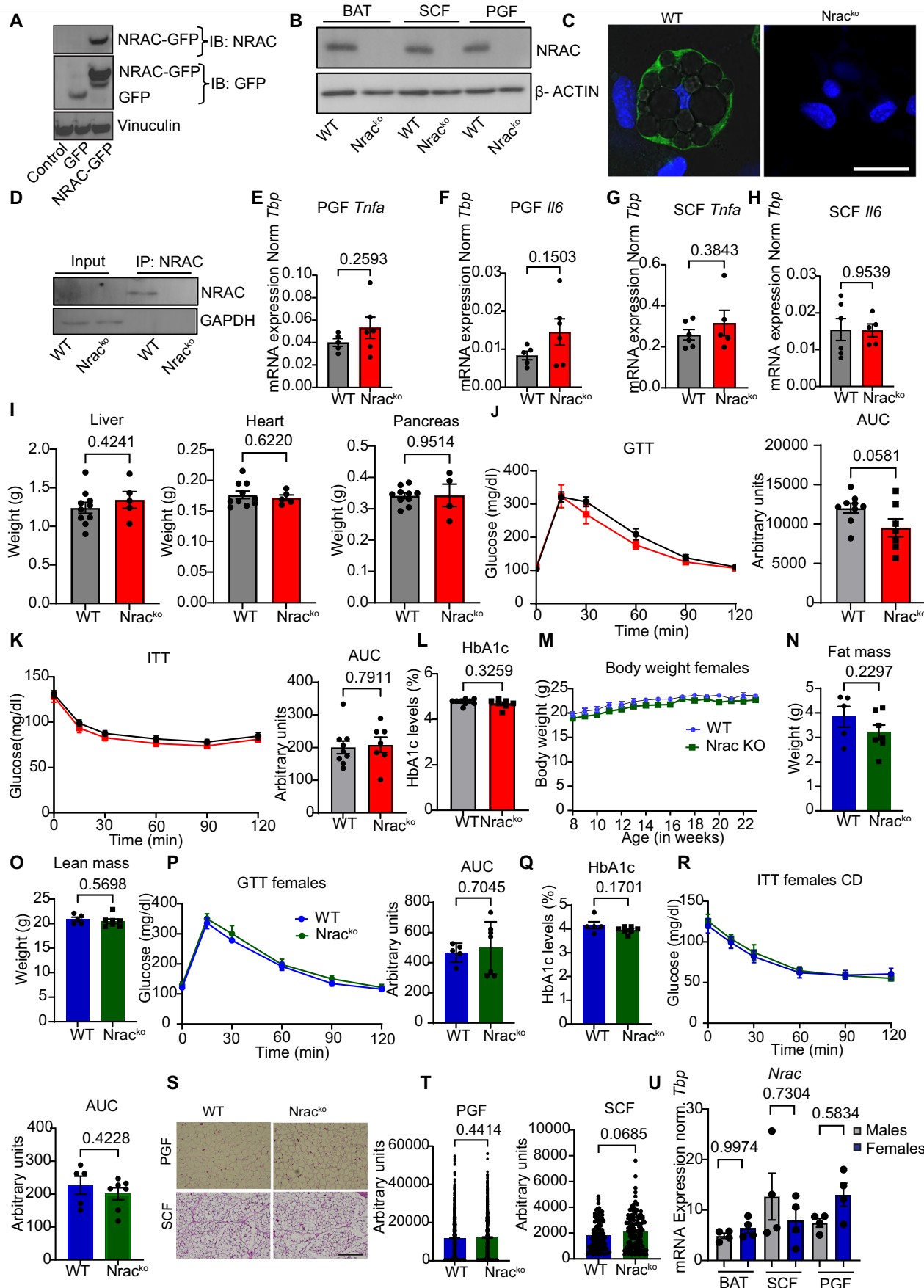

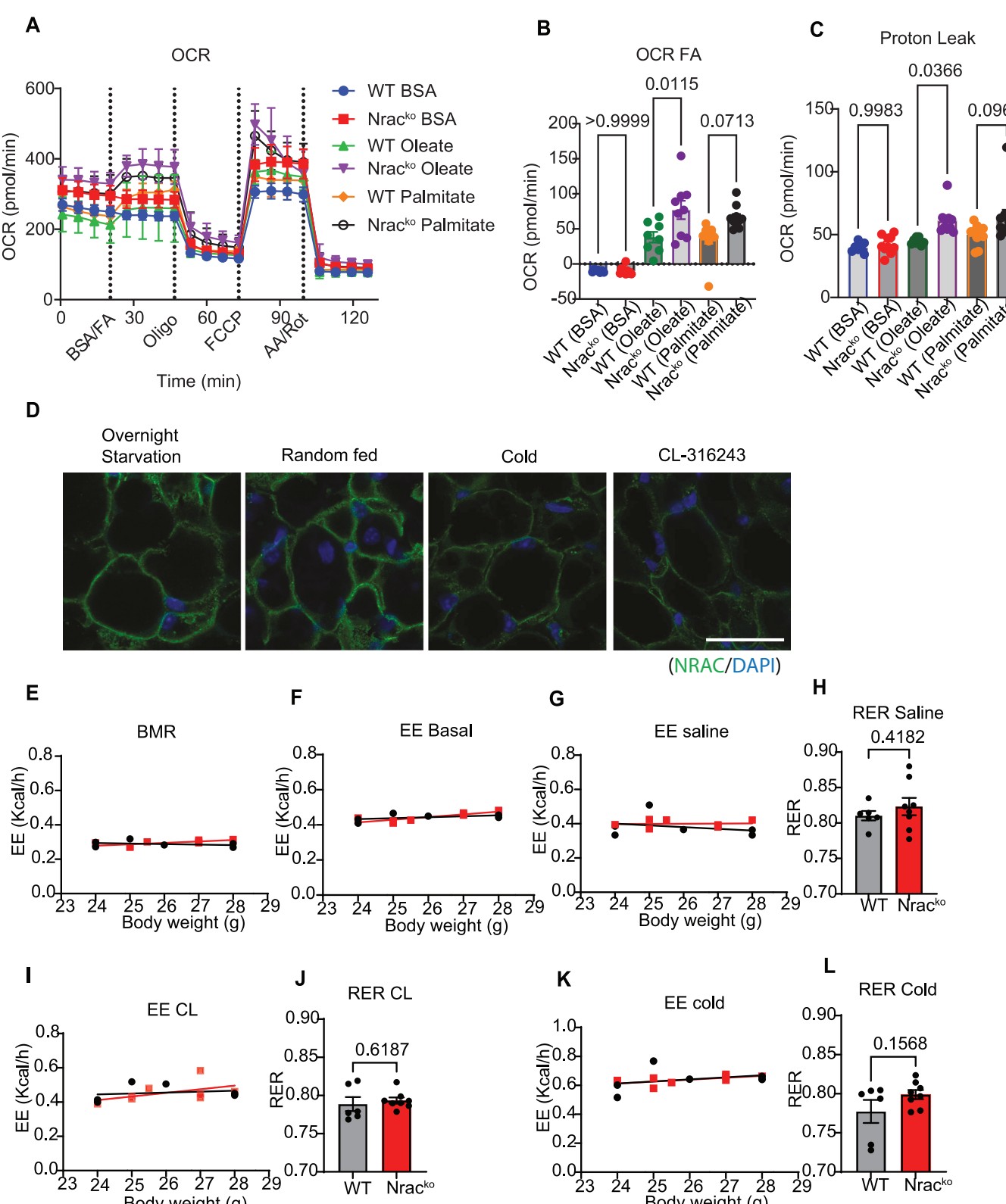

◀ **Figure EV2. NRAC deletion does not affect energy homeostasis, food intake and substrate utilization in mice under chow diet.**

(A) Oxygen consumption rates (OCR) in wt and Nrac$^{ko}$ differentiated adipocytes. Injection strategy included BSA or BSA conjugated oleate and palmitate (100 μM each) (port 1), oligomycin (port 2), FCCP (port 3) and Antimycin A and rotenone in port 4. (B) OCR calculated by subtracting OCR after fatty acid (FA) stimulation from basal respiration. (C) Proton leak calculated by subtracting oligomycin OCR from non-mitochondrial OCR ($n = 9$). (D) Immunostaining of BAT from wt mice either overnight starved (16 h), random fed, acute cold, (4 h) and CL-316243 injected mice ($n = 3$), age 12 weeks (scale 100 μm). (E) Basal metabolic rate (BMR). (F) Energy expenditure and respiratory exchange ratio (RER), (G, H) in saline treated mice, (I, J) in CL316243 treated mice and (K, L) upon acute cold exposure ($n = 6$ WT and 8KO, Age 16 weeks). Data are shown as mean ± SEM. *$P < 0.05$, **$P < 0.01$, ***$P < 0.001$ by Student's *t* test or one or two way ANOVA. In figures (E–G, I, K) the data are plotted as linear regression. Source data are available online for this figure.

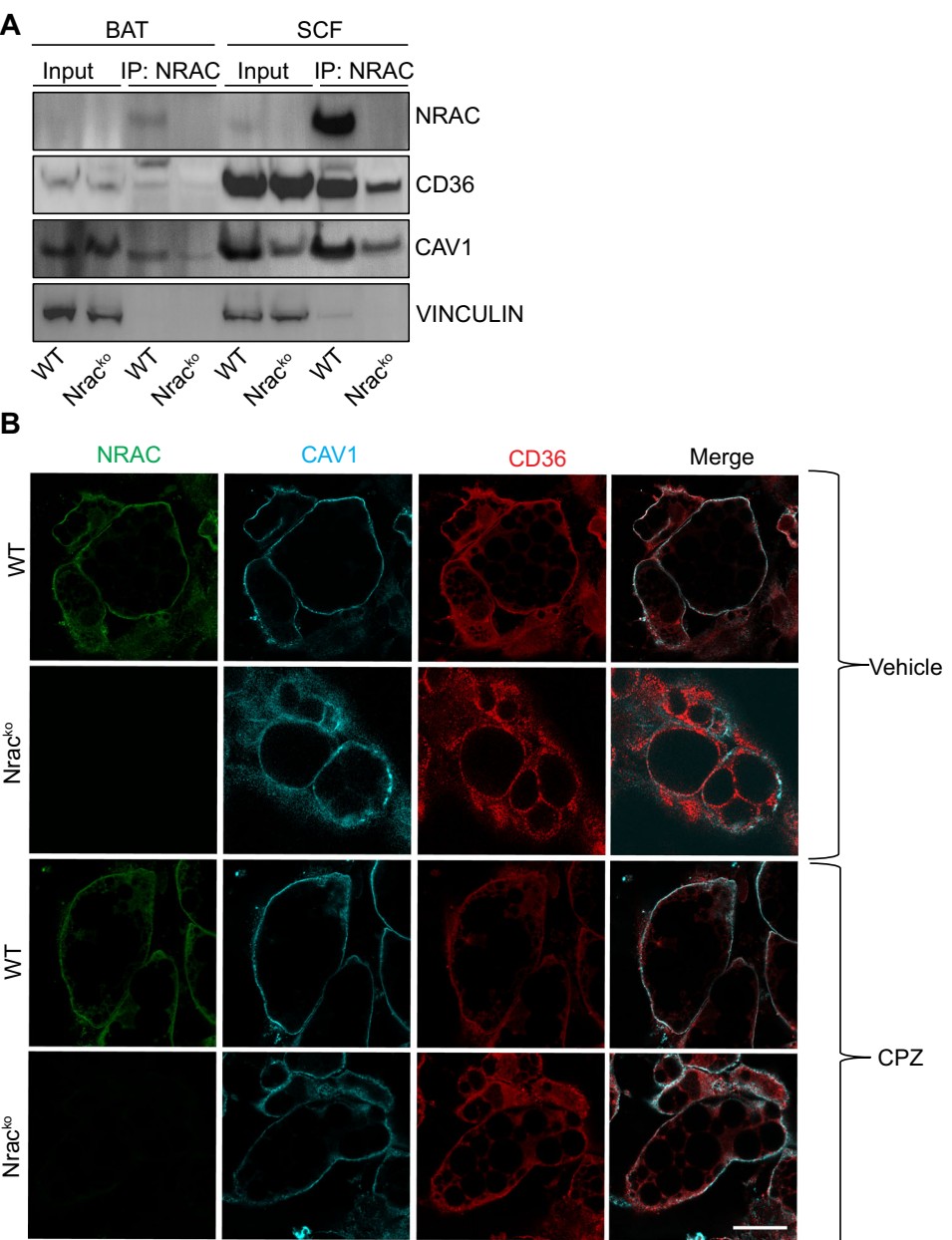

**Figure EV3. NRAC regulates CD36/Cav1 interaction in adipocytes.**

(A) Immunoprecipitation of NRAC from BAT and SCF. (B) Immunostaining of differentiated immortalized preadipocytes treated with vehicle (0.05% ethanol) or Chlorpromazine (CPZ) (10 μM) for 1 h (scale 25 μm). Source data are available online for this figure.

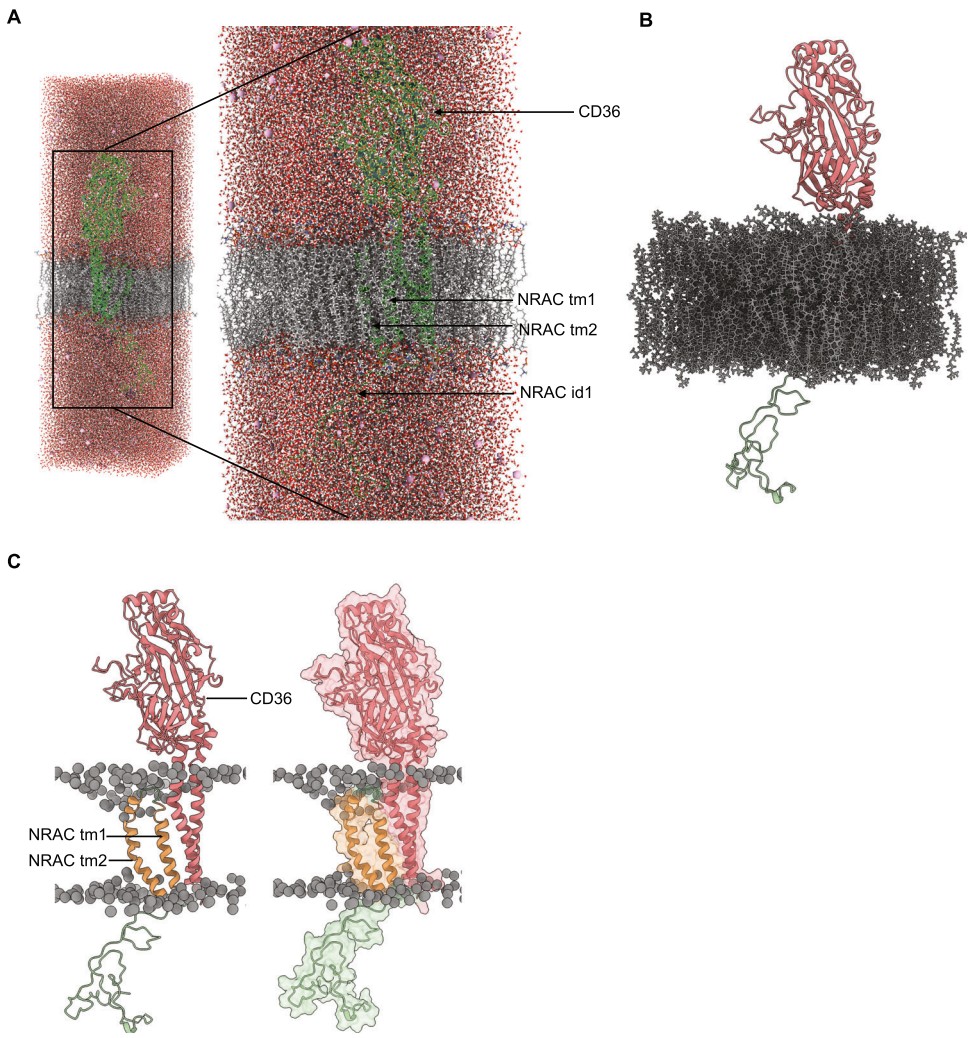

**Figure EV4. First transmembrane domain of NRAC is required for NRAC-CD36 interaction.**

(A) MD-simulations of human NRAC and CD36. The full simulation block is shown, with the central grey stripe representing the lipid bilayer membrane. Water molecules are depicted as red and white particles, while Na$^+$ and Cl$^-$ ions appear as pink spheres at a physiological concentration of 0.15 M. The bright green structures correspond to the proteins CD36 and NRAC, with CD36 embedded in the upper membrane leaflet and NRAC in the lower leaflet, and the zoomed in area of the same region. (B) CD36–NRAC complex is embedded in the membrane, with CD36 located on top and NRAC below. The membrane is shown in grey. (C) The CD36–NRAC complex is embedded in the membrane, with CD36 on top and NRAC below. Phospholipids in the membrane are represented as grey circles. Source data are available online for this figure.

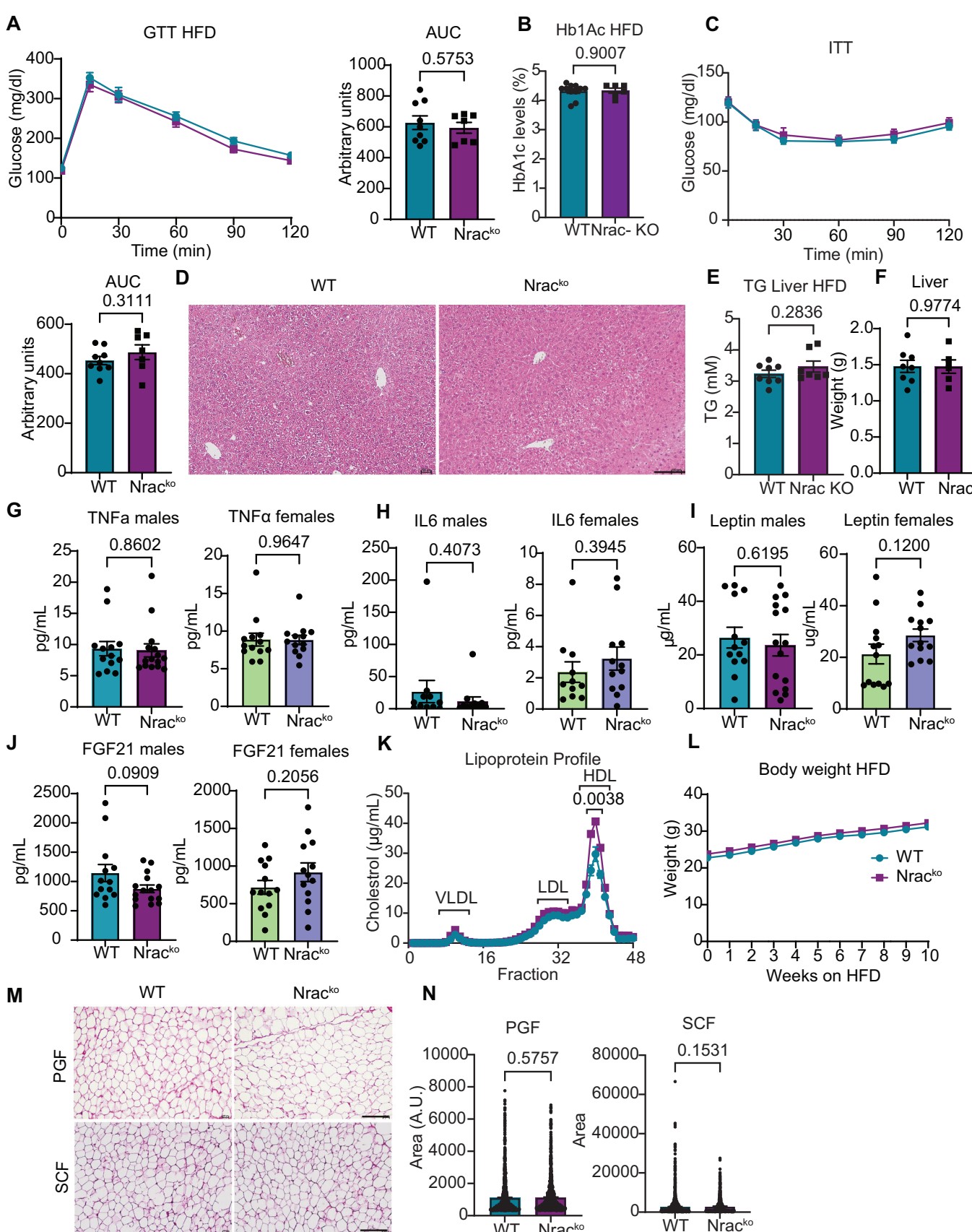

◄

**Figure EV5. HFD feeding does not affect glucose metabolism, insulin sensitivity and plasma profile of inflammatory cytokines in Nrac$^{ko}$ mice.**

(A) Glucose tolerance test and baseline corrected area under the curve ($n = 9$ wt, 7 ko, 10 weeks on HFD). (B) Percentage of HbA1c in mice13 weeks on HFD ($n = 9$ wt, 7 ko). (C) Insulin tolerance test and baseline corrected area under the curve, (12 weeks on HFD, $n = 9$ wt, 7 ko). (D) H&E staining of livers of HFD fed mice (17 weeks on HFD), (E) Liver triglyceride levels and (F) liver weight ($n = 9$ wt, 7 ko, 17 weeks on HFD), Plasma levels of (G) TNFα, (H) IL6, (I) Leptin, and (J) FGF21 in male and female non-fasted mice (male: 14 wt, 13 ko; female: 13 wt, 15 ko, 20 weeks on HFD). (K) Plasma lipoprotein profile ($n = 5$ wt, 5ko, 10 weeks on HFD), (L) Body weight development in HFD fed animals ($n = 8$ wt, 7 ko). Week 0 was the day when the diet was switched from chow to HFD (age 6 weeks). (M) H&E staining of PGF and SCF from HFD fed mice (17 weeks on HFD), and (N) the quantification of adipocyte cell size. Significance levels are indicated as follows: *$P < 0.05$, **$P < 0.01$, and ***$P < 0.001$. Data are shown as mean ± SEM. *$P < 0.05$, **$P < 0.01$, ***$P < 0.001$ by Student's *t* test in all figures except (A, C, K, L) where two-way ANOVA was used. Source data are available online for this figure.

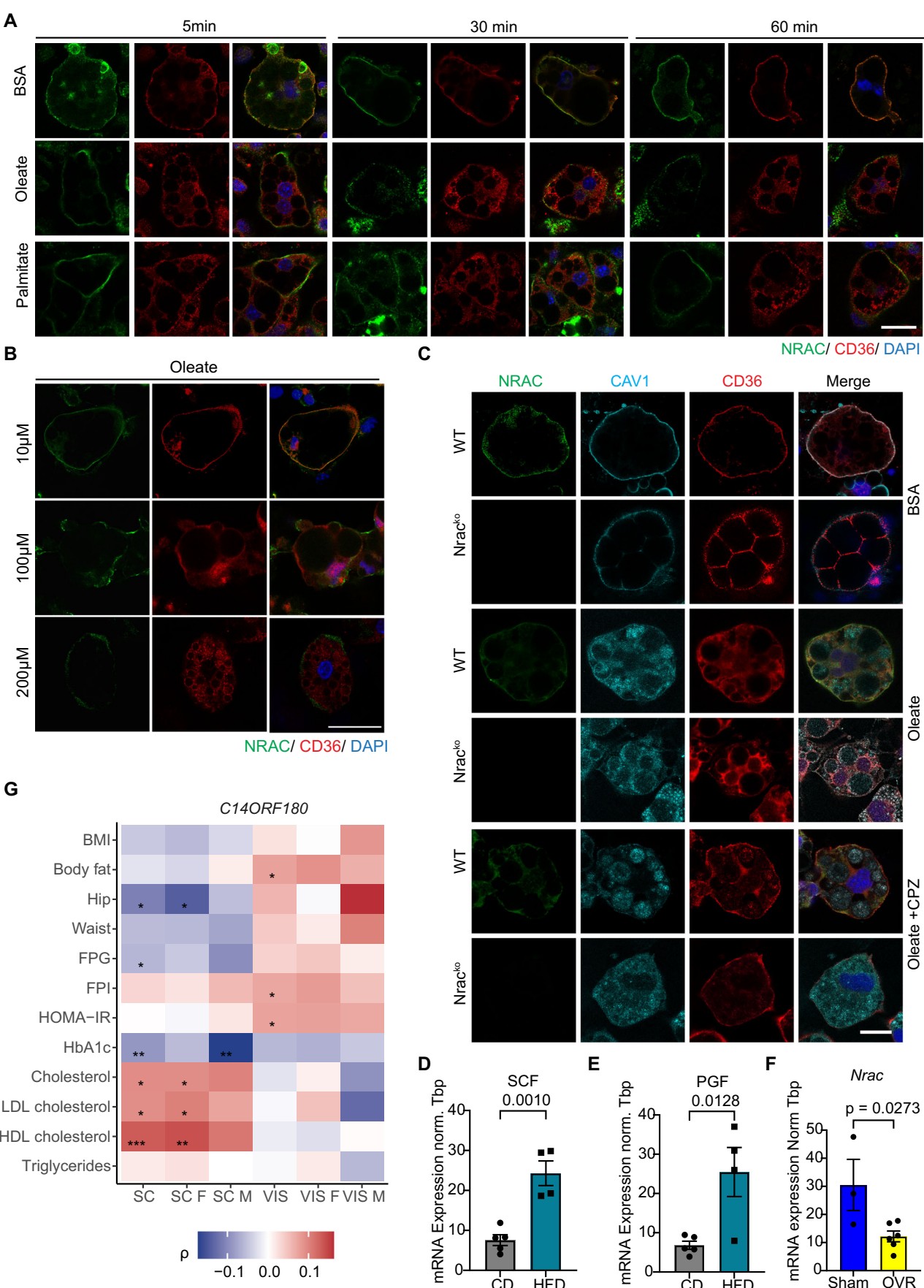

◀ **Figure EV6. Fatty acid stimulation reduces NRAC surface localization in adipocytes.**

(A) Immunostaining of differentiated subcutaneous adipocytes, treated with BSA, Oleate (100 μM) and palmitate (100 μM) for 5, 30 and 60 min. On day 8 of differentiation, cells were starved with serum free medium for 1 h and treated with fatty acids for the indicated times. (B) Immunostaining of differentiated subcutaneous adipocytes on day 8 of differentiation. Cells were starved with serum free medium 1 h and treated with 10, 100 and 200 μM BSA conjugated oleate. (C) Immunostaining of differentiated adipocytes treated either with oleate (100 μM) or with oleate (100 μM) and CPZ (10 μM) for one hour after 1 h serum starvation. *Nrac* mRNA expression in (D) subcutaneous (SCF) and (E) perigonadal (PGF) adipose tissue from CD and HFD fed mice ($n = 5$ CD and 4 HFD), (F) *Nrac* mRNA expression of sham and ovariectomized (OVR) HFD fed mice ($n = 3$ sham, 6 ovariectomized). (G) The correlation of C14orf180 (human *NRAC*) gene expression with clinical parameters was analyzed in both subcutaneous (SC) and visceral (VIS) adipose tissue across all individuals, as well as separately for male (M) and female (F) participants. Spearman correlations were calculated, and *P* values were adjusted for false discovery rate. Significance levels are indicated as follows: *$P < 0.05$, **$P < 0.01$, and ***$P < 0.001$. Data are shown as mean ± SEM. *$P < 0.05$, **$P < 0.01$, ***$P < 0.001$ by Student's *t* test in all figures except (A, C, K, L) where two-way ANOVA was used. Source data are available online for this figure.

