## [Peer Review File · The EMBO Journal]

NRAC controls CD36-mediated fatty acid uptake in adipocytes and lipid clearance *in vivo*

Inderjeet Singh, Yasuhiro Onogi, Filipe Menezes, Dina Khasanova, Lingru Kang, Chenxi Wang, Julio Ruiz Trave, Sapna Sharma, Ahmed Khalil, Valentin Reichenbach, Yingzi Shi, Andrew Flatley, Xiaocheng Yan, Andreas Israel, Nathalia Dragano, Juan Aguilar-Pimentel, Anne Hoffmann, Adhideb Ghosh, Falko Noé, Christian Wolfrum, Sebastian Cucuruz, Ann-Christine König, Ingo Bartscher, Stefanie Hauck, Heiko Lickert, Susanna Hofmann, Regina Feederle, Sonja Schriever, Rene Hernandez-Bautista, Gencer Sancar, Alberto Cebrian-Serrano, Igor Tetko, Helmut Fuchs, Valérie Gailus-Durner, Matthias Blüher, Martin Hrabé de Angelis, and Siegfried Ussar

Corresponding author: Siegfried Ussar (siegfried.ussar@helmholtz-munich.de)

Review Timeline:

Submission Date:	4th Nov 24
Editorial Decision:	13th Dec 24
Appeal:	30th May 25
Editorial Decision:	25th Jun 25
Revision Received:	2nd Jul 25
Accepted:	17th Jul 25

Editor: Daniel Klimmeck

Transaction Report:

Dear Dr. Ussar,

Thank you again for the submission of your manuscript (EMBOJ-2024-119521-T) to The EMBO Journal, as well as for your patience with our response. Your manuscript has been sent to three referees for evaluation with expertise in adipocyte biology and lipid metabolism, and we have received reports from all of them, which I enclose below. I regret to say that in light of their input we decided that we cannot offer publication in The EMBO Journal. We do encourage you to consider transfer to our sister journal EMBO Reports.

As you will see, the referees appreciate the potential interest of your results, identifying a novel fatty acid uptake mechanism in adipose cells, introducing Nrac as a new control factor of CD36 mediated lipid uptake in the context of caveolin-dependent endocytosis. However, they also raise major concerns with the analysis that I am afraid preclude publication here. In more detail, referee #2 points to important remaining gaps in the understanding provided regarding upstream control and reasons for the observed gender disparity (ref#2, pt.4; see also ref#3, pt.2). Referee #1 agrees in that more exploration of physiological upstream triggers of Nrac surface localisation are required (ref#1, pt.4). Reviewer #2 in addition expresses reservations regarding the discrepancies of current data with published results (ref#2, pt.5) and human relevance of the data (ref#2, pt.11). Reviewer #3 points to insufficient insights provided into the molecular details of how Nrac regulates CD36 complex formation and endocytosis (ref#3, pt.1; see also ref#1, pt.5).

Given these overall critical opinions from good experts on the field - please note the single major round of revisions we usually offer as to our journal policy - I am afraid we conclude that we cannot offer to publish your work at our venue.

That said, I discussed your manuscript and the referee reports with my colleague Deniz Senyilmaz Tiebe at our sister journal EMBO Reports. I am happy to share that Deniz finds the proposed role for NRAC in regulation of CD36 endocytosis is in principle very interesting. She states: 'I would be happy to invite a revised manuscript to EMBO Reports as outlined below:

- Additional insight into regulation of Nrac by physiological inputs is required - e.g. whether it is regulated by HFD (referee #3, point 4), how it is regulated by exogenous oleate (referee #3, point 5).
- Other concerns of referee #1 and #2 need to be satisfactorily addressed.
- Some more insight into the interaction between CD36 and NRAC e.g. by domain mapping is necessary (referee #3, point 1). I agree with the referee that structural insight would further strengthen the manuscript, but it is not required for EMBO Reports.
- Additional discussion points on the sexual dimorphic phenotypes of the NRAC ko mice and potential reasons explaining them need to be added into the manuscript (referee #2, point related to Fig S1; referee #3, point 2). Further experimental investigation of e.g. sexual hormones is not required for EMBO Reports although I agree with the referee that they would strengthen the work.

It is not necessary to revise the manuscript prior to transfer. Should that be of interest, the authors can transfer the manuscript as is. I would then send a formal decision letter inviting the authors to submit a revised version, and specifying the scope of the revision as above. I am happy to discuss any point regarding the revision with the authors.'

Thank you in any case for the opportunity to consider this manuscript. I regret we cannot be more positive on this occasion but hope nevertheless that you will find our referees' comments helpful and are favourable towards our transfer option.

Best regards,

Daniel Klimmeck

Daniel Klimmeck, PhD
Senior Editor
The EMBO Journal

Referee #1:

General summary

Whole-body Nrac deletion increases white adipocyte hypertrophy by enhancing fatty acid (FA) uptake from the circulation. Despite these changes, Nrac KO mice display no significant phenotypic differences compared to WT mice, and their systemic glucose metabolism remains healthy. High-fat diets eliminate these differences, suggesting limited physiological relevance under

normal conditions.

Mechanistically, Nrac forms a ternary complex with CD36 and caveolin-1, facilitating their interaction. Nrac loss disrupts this interaction, promoting CD36 internalization via clathrin-mediated endocytosis, leading to adipocyte hypertrophy under low extracellular FA conditions. However, these effects do not affect systemic glucose homeostasis or fat metabolism.

Specific major concerns essential to be addressed to support the conclusions

(1) Since AA/Rot treatment does not bring the OCR to zero (Fig. 2E), all OCR values should subtract the AA/Rot values to quantify them more accurately in the bar graphs of 2F-H. One could argue that the non-stimulated OCR could also be subtracted from the stimulated OCR to quantify ISO OCR, leak OCR, and max OCR with more accuracy.

(2) In Fig. 2E, at least half of the ISO-induced respiration is inhibited by oligomycin, suggesting that there is a UCP1-independent thermogenic component, and this should be quantified as well as a bar graph.

(3) Nrac KO white and brown adipocytes exhibit increased fatty acid (FA) influx. However, this leads to differing outcomes: white adipocytes become larger, while brown adipocytes become smaller. The argument that brown adipocytes are smaller due to increased UCP1 activity is not really supported, because FA influx was measured under non-stimulated conditions (i.e., not ISO-induced). The ISO-induced condition is necessary to observe differences in thermogenesis between wild-type (WT) and Nrac KO brown adipocytes, as shown in Fig. 2E. Basal thermogenesis does not differ between WT and Nrac KO brown adipocytes (Fig. 2E). Therefore, what could explain the increased FA uptake yet reduced fat storage in brown adipocytes?

(4) The authors hypothesize, based on their in vitro data, that high extracellular FA levels reduce Nrac surface localization. Can the authors test this in vivo with cold, fasting, or CL 316,243 treatment? They could monitor white and brown adipose depots to see if these stimuli that increase systemic FA concentrations influence Nrac localization in vivo.

(5) Do circulating FAs alter Nrac mRNA or protein expression? Or is just CD36 binding that is affected?

(6) What consequence, if any, does Nrac overexpression have on FA uptake?

Referee #2:

Comments to the manuscript EMBOJ-2024-119521-T:

The manuscript entitled "Nrac regulates CD36 mediated fatty acid uptake in adipocytes" by Singh et al. investigates the role of Nutritionally regulated adipose and cardiac enriched protein (Nrac) in fatty acids (FAs) uptake in adipocytes. The authors show that Nrac modulates CD36-mediated FAs uptake by forming a complex with it and caveolin-1 (CAV-1) when the concentration of extracellular FAs is low. Depletion of Nrac or increased concentration of extracellular FAs, leading to reduced Nrac cell surface localisation, leads to CD36 dissociation from CAV-1, and its internalisation via Clathrin-mediated endocytosis. This increases FAs uptake into adipocytes, adipocyte hypertrophy, increased fat mass and elevated lipid clearance from blood in chow fed mice.

The novelty of this work resides in unravelling a new regulatory mechanism of adipocytes FAs uptake dependent on their extracellular availability.

Major comments

1) Fig.S1B. In this figure the indications of which samples in the immunoblot are from KO or WT mice are missing.

2) Fig.1. In this figure, the authors show that the KO mice present increased total fat and adipose tissue depots mass, but no difference in body weight vs WT controls. Is any of the other organs of the KO mice lighter? Since the body weight is the same there should be some difference in one or some other organs weight to compensate and end up with similar body weight between KO and WT mice. The authors should provide this information.

3) Fig.1H. Referring to this figure, the authors mention that they did not see sign of increased immune cells infiltration, however there is no indication of measurements specifically regarding immune cells there. The authors should correct their statement.

4) Fig.S1. Males and females KO mice vs controls present clearly gender dysmorphism in their phenotype. The authors have any idea why that happen? Is Nrac regulated by sex hormones?

5) Fig.1M. Do the authors have any explanation regarding the discrepancy they observed between their result regarding the differentiation of the primary adipocyte of their KO mouse (i.e. not difference vs control) and the results of the other papers (Kerr

et al, 2019 and Zhang et al, 2012) i.e Nrac depletion impairing adipocyte differentiation? Differences in the culture media composition?

6) Fig.2C. The authors should provide a quantification of the immunoblot.

7) Fig.3A and B. The authors should provide these same data also for SCF and BAT of the KO mice vs controls.

8) Fig.3B. In this immunoblot of the IP, the difference between the signal of CD36 binding Nrac in WT and KO mice is barely visible. The authors should provide a more representative immunoblot where the difference is more visible.

9) Fig.S4K. The authors show a significant increase in HDL in KO mice vs controls. Do the authors have an explanation for that phenotype?

10) Fig. S4P and Fig.4I. Have the authors an explanation for the discrepancy between Nrac mRNA and protein levels upon HFD feeding in PGF and SCF of WT mice?

11) Fig.4G. We do not see the relevance of the human obese subject's data provided with respect to strengthen the work described in the paper.

Referee #3:

This paper presents an interesting and comprehensive study on the role of Nrac (Nutritionally regulated adipose and cardiac enriched protein) in regulating fatty acid uptake in adipocytes. The authors provide evidence that Nrac acts as a rheostat for CD36-mediated fatty acid uptake by controlling the endocytic pathway of CD36 depending on extracellular fatty acid concentrations. The authors present a novel mechanism for regulating fatty acid uptake in adipocytes, which could have important implications for understanding obesity and related metabolic disorders. The paper is well-structured, with a logical flow from in vivo phenotyping to mechanistic studies and human data correlation. The figures are generally clear and informative, presenting a large amount of data in a digestible format. In conclusion, this manuscript presents novel and important findings on the role of Nrac in regulating fatty acid uptake in adipocytes. Addressing the below points could further strengthen the paper and increase its impact in the field of adipocyte biology and metabolism.

1. While the authors propose that Nrac forms a ternary complex with CD36 and caveolin-1, the exact molecular mechanism of how Nrac regulates this interaction is not fully elucidated. Additional experiments, such as domain mapping or structural studies, could provide more insight into this process.
2. The authors observe phenotypic differences between male and female Nrac knockout mice on chow diet. However, they do not explore potential reasons for these differences or discuss their implications. Further investigation into the hormonal or other factors contributing to these differences could strengthen the paper.
3. While the authors show increased UCP1 expression and oxygen consumption in Nrac knockout BAT, they do not observe changes in whole-body energy expenditure. This discrepancy could be further investigated or discussed in more detail. Similarly, the lack of phenotypic differences between wild-type and Nrac knockout mice on HFD is intriguing but not fully explained.
4. Is Nrac regulated by any physiological contexts? The authors show that HFD mimics the switch in endocytotic mechanisms, is Nrac regulated by HFD for example?
5. The oleate-induced reduction in Nrac surface expression could be further investigated. How rapidly does this occur? How specific is this to the oleate lipid moiety? Is there dose dependency of oleate concentration extracellularly?

offer to publish to another EMBO publication or the open access journal Life Science Alliance launched in partnership between EMBO Press, Rockefeller University Press and Cold Spring Harbor Laboratory Press. The full manuscript and if applicable, reviewers' reports, are automatically sent to the receiving journal to allow for fast handling and a prompt decision on your manuscript. For more details of this service, and to transfer your manuscript please click on Link Not Available. **

Referee #1:

General Summer

Whole-body Nrac deletion increases white adipocyte hypertrophy by enhancing fatty acid (FA) uptake from the circulation. Despite these changes, Nrac KO mice display no significant phenotypic differences compared to WT mice, and their systemic glucose metabolism remains healthy. High-fat diets eliminate these differences, suggesting limited physiological relevance under normal conditions.

Thank you very much for your careful assessment of our work. We would like to point out, however, that the absence of a phenotype upon HFD feeding is, in our opinion, a real strength of our manuscript as we show that high fatty acid concentrations result in proteasomal degradation of NRAC, increasing fatty acid uptake into adipocytes. Hence, HFD feeding and genetic knockout of NRAC produce a similar phenotype. Thus, on HFD, wildtype mice essentially behave like NRAC knockout mice.

Mechanistically, Nrac forms a ternary complex with CD36 and caveolin-1, facilitating their interaction. Nrac loss disrupts this interaction, promoting CD36 internalization via clathrin-mediated endocytosis, leading to adipocyte hypertrophy under low extracellular FA conditions. However, these effects do not affect systemic glucose homeostasis or fat metabolism.

We now include a much more in depth analysis of the molecular mechanisms underlying the NRAC/CD36 interaction and provide data that high extracellular fatty acid concentrations result in ubiquitination and proteasomal degradation of NRAC. As wildtype mice also loose NRAC upon HFD feeding we cannot conclude that NRAC does not impact on systemic glucose homeostasis. Future experiments with degradation-resistant NRAC mutants will be necessary to answer this question. However, identifying the key amino acids, generating the mutant cell and mouse lines and analysing them will take several years, which we hope the reviewer agrees should be presented in a subsequent manuscript.

Specific major concerns essential to be addressed to support the conclusions

(1) Since AA/Rot treatment does not bring the OCR to zero (Fig. 2E), all OCR values should subtract the AA/Rot values to quantify them more accurately in the bar graphs of 2F-H. One could argue that the non-stimulated OCR could also be subtracted from the stimulated OCR to quantify ISO OCR, leak OCR, and max OCR with more accuracy.

We thank the reviewer for this insightful comment and have reanalysed all data. We have calculated the proton leak by subtracting Oligomycin OCR from non-mitochondrial respiration, and maximal respiration was calculated by subtracting data from FCCP OCR from non-mitochondrial respiration.

The ISO OCR was calculated by subtracting basal (non-stimulated) OCR from isoproterenol OCR. The reserve capacity was also calculated by subtracting FCCP OCR from basal OCR (Fig 2G-K). We have updated the figures and text accordingly (see also below).

(2) In Fig. 2E, at least half of the ISO-induced respiration is inhibited by oligomycin, suggesting that there is a UCP1-independent thermogenic component, and this should be quantified as well as a bar graph.

We now quantified ATP-linked respiration by measuring the oligomycin-sensitive fraction of the ISO-stimulated oxygen consumption rate (OCR). While the most direct approach to assess UCP1-independent thermogenesis involves using UCP1 knockout models, we included a bar graph in the revised Figure 2L to illustrate ATP-

linked respiration. This highlights the oligomycin-sensitive component, which may serve as an indicator of UCP1-independent thermogenic activity.

(3) Nrac KO white and brown adipocytes exhibit increased fatty acid (FA) influx. However, this leads to differing outcomes: white adipocytes become larger, while brown adipocytes become smaller. The argument that brown adipocytes are smaller due to increased UCP1 activity is not really supported, because FA influx was measured under non-stimulated conditions (i.e., not ISO-induced).

We thank the reviewer for this valuable comment. To address this point, we have now measured fatty acid uptake in brown adipocytes under both non-stimulated (DMSO) and isoproterenol-stimulated conditions. In both cases, Nrac KO brown adipocytes exhibited increased FA uptake compared to WT, indicating that the enhanced influx is not limited to basal conditions but also occurs upon β -adrenergic stimulation. We have updated the figure and text accordingly (see also below).

(4) The ISO-induced condition is necessary to observe differences in thermogenesis between wild-type (WT) and Nrac KO brown adipocytes, as shown in Fig. 2E. Basal thermogenesis does not differ between WT and Nrac KO brown adipocytes (Fig. 2E). Therefore, what could explain the increased FA uptake yet reduced fat storage in brown adipocytes?

We thank the reviewer for this important point. As noted, basal thermogenesis in Fig. 2E was measured under fatty acid-free conditions, which may not fully reflect the metabolic potential of brown adipocytes. To address this, we performed additional experiments in which exogenous fatty acids were injected into the Seahorse assay system. Under these conditions, Nrac KO brown adipocytes exhibited a significantly

higher oxygen consumption rate (OCR) compared to WT, suggesting enhanced oxidative metabolism in the presence of fatty acids.

These findings support our conclusion that the increased fatty acid uptake observed in Nrac KO brown adipocytes is functionally coupled to elevated mitochondrial respiration, rather than lipid storage. This may explain the reduced lipid accumulation despite increased FA influx. We have included these new data in the revised manuscript and updated Figure (Fig 2M-O and S2I-K) accordingly.

(5) The authors hypothesize, based on their in vitro data, that high extracellular FA levels reduce Nrac surface localization. Can the authors test this in vivo with cold, fasting, or CL 316,243 treatment? They could monitor white and brown adipose depots to see if these stimuli that increase systemic FA concentrations influence Nrac localization in vivo.

Thank you very much for raising this interesting point. Indeed we have shown in our previous version that HFD feeding reduces NRAC localisation in white adipocytes. We now tested NRAC expression/localisation in brown adipocytes in vivo upon overnight starvation, cold exposure and acute CL-316243 treatment. In contrast to HFD feeding, these stimuli only had modest effects on NRAC localisation. We now

show these data as Fig. S2D (see below). Overall, these data indicate that the extracellular FA concentrations required for NRAC degradation will require additional in vivo experiments, which we could not get approved by the local government in time for this revisions.

(6) Do circulating FAs alter Nrac mRNA or protein expression? Or is just CD36 binding that is affected?

We previously showed that HFD feeding increases Nrac expression in SCF and PGF, however results in reduced surface localization of Nrac. We now show data that fatty acid stimulation leads to Nrac ubiquitination, and proteasomal degradation (Fig 5I). Hence, the rapid dissociation of CD36 from caveolin-1 upon high fatty acid stimulation appears to be due to proteasomal degradation of NRAC. This further indicates that the observed increase in mRNA expression could be compensatory.

(7) What consequence, if any, does Nrac overexpression have on FA uptake?

Stable overexpression of Nrac in wt and Nrac^{ko} adipocytes significantly reduced fatty acid uptake in both wt and Nrac ko adipocytes compared to the controls (new Fig. 1K). Moreover, transient overexpression of full length Nrac (Nrac 1-165) in HEK293T cells resulted in decreased fatty acid uptake. Importantly, we demonstrate that the interaction between Nrac and CD36 is essential for this inhibitory effect. When a Nrac construct lacking the first transmembrane domain (abolishing

NRAC/CD36 interaction) was overexpressed, fatty acid uptake remained unchanged (new Fig 4C).

Referee #2:

Comments to the manuscript EMBOJ-2024-119521-T:

The manuscript entitled "Nrac regulates CD36 mediated fatty acid uptake in adipocytes" by Singh et al. investigates the role of Nutritionally regulated adipose and cardiac enriched protein (Nrac) in fatty acids (FAs) uptake in adipocytes. The authors show that Nrac modulates CD36-mediated FAs uptake by forming a complex with it and caveolin-1 (CAV-1) when the concentration of extracellular FAs is low. Depletion of Nrac or increased concentration of extracellular FAs, leading to reduced Nrac cell surface localisation, leads to CD36 dissociation from CAV-1, and its internalisation via Clathrin-mediated endocytosis. This increases FAs uptake into adipocytes, adipocyte hypertrophy, increased fat mass and elevated lipid clearance from blood in chow fed mice.

The novelty of this work resides in unravelling a new regulatory mechanism of adipocytes FAs uptake dependent on their extracellular availability.

We would like to thank the reviewer for her/his careful assessment of our manuscript.

Major comments

1) Fig.S1B. In this figure the indications of which samples in the immunoblot are from KO or WT mice are missing.

We would like to apologize for this error and have corrected the mistake. We replaced the image with a clearer blot (new Figure S1B see below).

2) Fig.1. In this figure, the authors show that the KO mice present increased total fat and adipose tissue depots mass, but no difference in body weight vs WT controls. Is any of the other organs of the KO mice lighter? Since the body weight is the same there should be some difference in one or some other organs weight to compensate and end up with similar body weight between KO and WT mice. The authors should provide this information.

We did not observe significant differences in the weights of other organs (new Fig S1I and see below). We would like to clarify that we saw a difference in fat mass of around 1g between wt and ko mice. Thus, the reason why we did not observe differences in total body weight is most likely due to the fact that this relatively small difference becomes not significant in context of the whole body weight.

3) Fig.1H. Referring to this figure, the authors mention that they did not see sign of increased immune cells infiltration, however there is no indication of measurements specifically regarding immune cells there. The authors should correct their statement.

We thank the reviewer for raising this important point. We referred to the absence of any crown-like structures in the H&E stainings from adipose depots of wt and ko mice. We have now also included qPCR data fo TNFa and IL6 as Figure S1E-H (see also below).

4) Fig.S1. Males and females KO mice vs controls present clearly gender dysmorphism in their phenotype. The authors have any idea why that happen? Is Nrac regulated by sex hormones?

Thank you very much for giving us the opportunity to clarify this important point. We now included various additional experiments to address this. Most importantly, we show that both the lipid tolerance test and the adipocyte fatty uptake assay confirm also increased fatty acid uptake in female Nrac knockout mice. Hence, on a molecular and cell biological level there is not gender dysmorphism. However, the reviewer is absolutely right that the consequences of NRAC deletion on adipocyte hypertrophy are absent in female mice. However, female mice are in general

protected from diet induced obesity and the development of insulin resistance due to increased systemic energy expenditure.

Moreover, we now also compared Nrac mRNA expression between female and male mice, which did not reveal major differences. However we find that HFD feeding increases Nrac expression in both male and female mice and HFD ovariectomized females showed decreased expression of Nrac in perigonadal adipose tissue compared to sham operated controls. Thus, sex hormones appear to impact on Nrac expression. However, it remains to be determined if this is a direct or indirect action. In summary, we now provide data that the effect of NRAC on adipocyte fatty acid uptake is the same between male and female mice. However, the physiological consequences appear to be dominated by differences in energy expenditure between sexes.

5) Fig.1M. Do the authors have any explanation regarding the discrepancy they observed between their result regarding the differentiation of the primary adipocyte of their KO mouse (i.e. not difference vs control) and the results of the other papers (Kerr et al, 2019 and Zhang et al, 2012) i.e Nrac depletion impairing adipocyte differentiation? Differences in the culture media composition?

Thank you for your question regarding the discrepancy between our findings and those reported by Kerr et al. (2019). We believe that the key difference lies in the experimental models used. Kerr et al. employed human mesenchymal stem cells (MSCs) and transiently knocked down Nrac using siRNA. In contrast, our study utilized primary preadipocytes isolated directly from WT and Nrac KO mice, which were then differentiated into mature adipocytes. A more detailed analysis would require us to obtain these cell lines and siRNAs, which we believe would not add additional information to our current manuscript.

6) Fig.2C. The authors should provide a quantification of the immunoblot.

We have now quantified the blot Figure 2D-E.

7) Fig.3A and B. The authors should provide these same data also for SCF and BAT of the KO mice vs controls.

We have now performed the co-IP with brown and subcutaneous adipose tissue from WT and Nrac^{ko} mice (Figure S3A).

8) Fig.3B. In this immunoblot of the IP, the difference between the signal of CD36 binding Nrac in WT and KO mice is barely visible. The authors should provide a more representative immunoblot where the difference is more visible.

We have now replaced the immunoblot of the IP with a more representative one to better illustrate our conclusions. The blot is shown in Figure 3B as below.

9) Fig.S4K. The authors show a significant increase in HDL in KO mice vs controls. Do the authors have an explanation for that phenotype?

Thank you for your observation regarding the increase in HDL levels in KO mice (Fig. S4K). This phenotype aligns with previous findings showing that brown adipose tissue (BAT) activation, such as through cold exposure, enhances HDL production. As demonstrated by Bartelt et al. (2011, <https://doi.org/10.1038/nm.2297>), this occurs via a reduction in circulating triglyceride-rich lipoprotein (TRL) particles and an increase in TRL-derived HDL precursors. We included these data to further support our conclusion of increased brown adipocyte activity.

10) Fig. S4P and Fig.4I. Have the authors an explanation for the discrepancy between Nr3c1 mRNA and protein levels upon HFD feeding in PGF and SCF of WT mice?

Our previous and new data show that upon increased extracellular fatty acid concentration NR3C1 undergoes ubiquitination and proteasomal degradation, which can explain the reduced protein abundance. To this end, we conclude that the increased mRNA expression is a cellular response to compensate the reduced protein activity, albeit experimental validation of the transcriptional control of Nr3c1 expression is still outstanding.

11) Fig.4G. We do not see the relevance of the human obese subject's data provided with respect to strengthen the work described in the paper.

Thank you very much for raising this important point. We agree with the reviewers that in principle we show a disconnect between mRNA expression and protein levels as described above. Our data would suggest that upregulation of Nrac mRNA expression, especially in obese subjects could reflect reduced protein function. During the revisions of this manuscript we tried to establish C14ORF180 stainings on human adipose tissue from lean and obese subjects. However, we did not succeed in obtaining convincing stainings, which precluded us from including these data into our manuscript. We discussed with the editor to exclude the data based on the reviewers comments. However, the recommendation was to retain this human data set. To further strengthen this part of our manuscript, we now included additional genomic analysis of SNPs within the C14orf180 locus and their association with obesity and metabolic traits. Indeed we could identify one intronic SNP that associates with obesity. At this point, however, as this is usually the case with reporting genomic or gene expression data, the impact of these observations on protein function remains to be determined.

Referee #3:

This paper presents an interesting and comprehensive study on the role of Nrac (Nutritionally regulated adipose and cardiac enriched protein) in regulating fatty acid uptake in adipocytes. The authors provide evidence that Nrac acts as a rheostat for CD36-mediated fatty acid uptake by controlling the endocytic pathway of CD36 depending on extracellular fatty acid concentrations. The authors present a novel mechanism for regulating fatty acid uptake in adipocytes, which could have important implications for understanding obesity and related metabolic disorders. The paper is well-structured, with a logical flow from in vivo phenotyping to mechanistic studies and human data correlation. The figures are generally clear and informative, presenting a large amount of data in a

digestible format. In conclusion, this manuscript presents novel and important findings on the role of Nrac in regulating fatty acid uptake in adipocytes. Addressing the below points could further strengthen the paper and increase its impact in the field of adipocyte biology and metabolism.

We would like to thank the reviewer for the very positive assessment of our work.

1. While the authors propose that Nrac forms a ternary complex with CD36 and caveolin-1, the exact molecular mechanism of how Nrac regulates this interaction is not fully elucidated. Additional experiments, such as domain mapping or structural studies, could provide more insight into this process.

To further investigate the molecular mechanism underlying the interaction between Nrac and CD36, we generated several deletion mutants of mouse Nrac and co-transfected them with CD36 in HEK293T cells (Fig 4A-B). Co-immunoprecipitation experiments revealed that the interaction between Nrac and CD36 is mediated through the first transmembrane (TM) domain of Nrac. Furthermore, molecular dynamics (MD) simulations supported this finding, independently predicting that the first TM domain of human Nrac is critical for its interaction with CD36 (Fig 4D and S4A-C). These results provide further mechanistic insight into how Nrac regulates the formation of the ternary complex. We hope with that these additional experiments can properly address the reviewers concern.

2. The authors observe phenotypic differences between male and female Nrac knockout mice on chow diet. However, they do not explore potential reasons for these differences or discuss their implications. Further investigation into the hormonal or other factors contributing to these differences could strengthen the paper.

Thank you very much for giving us the opportunity to clarify this important point. We now included various additional experiments to address this. Most importantly, we show that both the lipid tolerance test and the adipocyte fatty uptake assay confirm increased fatty acid uptake in female Nrac knockout mice. Hence, on a molecular and cell biological level there is not gender dysmorphism. However, the reviewer is absolutely right that the consequences of NRAC deletion on adipocyte hypertrophy are absent in female mice. However, female mice are in general protected from diet induced obesity and the development of insulin resistance due to increased systemic energy expenditure.

Moreover, we now also compared Nrac mRNA expression between female and male mice, which did not reveal major differences. However we find that HFD feeding increases Nrac expression in both male and female mice and HFD ovariectomized females showed decreased expression of Nrac in perigonadal adipose tissue compared to sham operated controls. Thus, sex hormones appear to impact on Nrac expression. However, it remains to be determined if this is a direct or indirect action. In summary, we now provide data that the effect of NRAC on adipocyte fatty acid uptake is the same between male and female mice. However, the physiological consequences appear to be dominated by differences in energy expenditure between sexes.

3. While the authors show increased UCP1 expression and oxygen consumption in Nrac knockout BAT, they do not observe changes in whole-body energy expenditure. This discrepancy could be further investigated or discussed in more detail.

We agree with the reviewer that the absence of measurable differences in whole-body energy expenditure are disappointing. There are several potential explanations for this. Due to limits in the amount of metabolic cages and the approved number of animals that can be tested in this experiment by the government of Upper Bavaria it could be that small differences in energy expenditure cannot be captured with statistical significance. Moreover, as shown in our H&E stainings adult BAT of Nrac ko mice shows a strong reduction in triglycerides. Recent data indicate that this, however, is important for thermogenesis independent of nutrient uptake. Thus, assessment of neonatal BAT activity would be necessary, which is not possible in metabolic cages. We commented on these potential explanations briefly, but did not want to include too many speculations in our manuscript, but rather describe our findings, with a focus on the role of NRAC in regulating CD36 uptake kinetics via regulation of its endocytic pathways.

4. Similarly, the lack of phenotypic differences between wild-type and Nrac knockout mice on HFD is intriguing but not fully explained. Is Nrac regulated by any physiological contexts? The authors show that HFD mimics the switch in endocytotic mechanisms, is Nrac regulated by HFD for example?

We show that HFD feeding increases Nrac mRNA expression in SCF and PGF, however results in reduced surface localization of Nrac. We now show data that fatty acid stimulation leads to Nrac ubiquitination, and proteasomal degradation (Fig 5I). Hence, the rapid dissociation of CD36 from caveolin-1 upon high fatty acid stimulation appears to be due to proteasomal degradation of NRAC. This further indicates that the observed increase in mRNA expression could be compensatory.

6. The oleate-induced reduction in Nrac surface expression could be further investigated. How rapidly does this occur? How specific is this to the oleate lipid moiety? Is there dose dependency of oleate concentration extracellularly?

Previous studies have shown that various fatty acids, including oleate, palmitate, and stearic acid, can induce internalization of CD36 (Lu et al., 2020). In line with these findings, we show that both oleate and palmitate result in increased CD36 internalization and reduce Nr3c1 surface localization. This effect is time and dose dependent, with prolonged exposure leading to a more pronounced reduction in surface expression (Figure S5O and P and see below). We hope this new data set can adequately address the reviewer's point.

Dear Dr. Ussar,

Thank you for submitting your revised manuscript (EMBOJ-2024-119521R-Q) to The EMBO Journal. As mentioned, based in our assessment of your rebuttal response, the amended study was sent back to the referees for their scientific reassessment, and we have received re-reports from two of them, which I enclose below. As you will see, the experts state that the work has been substantially enhanced by the revisions and they are now in favour of publication. Please note that while referee #1 was at this time not able to reassess your amended study, we have re-evaluated your response to his/her issues editorially and found these to be addressed satisfactorily.

Thus, we are pleased to inform you that your manuscript has been accepted in principle for publication in The EMBO Journal.

We now need you to take care of a number of issues related to formatting and data presentation as detailed below, which should be addressed at re-submission.

Please contact me at any time if you have additional questions related to below points.

As you might have noted on our webpage, every paper at the EMBO Journal now includes a 'Synopsis', displayed on the html and freely accessible to all readers. The synopsis includes a 'model' figure as well as 2-5 one-short-sentence bullet points that summarize the article. I would appreciate if you could provide this figure and the bullet points.

Thank you for giving us the chance to consider your manuscript for The EMBO Journal. I look forward to your final revision.

Again, please contact me at any time if you need any help or have further questions.

Best regards,

Daniel Klimmeck

>> Author Contributions: Remove the author contributions information from the manuscript text. Note that CRediT has replaced the traditional author contributions section as of now because it offers a systematic machine-readable author contributions format that allows for more effective research assessment. and use the free text boxes beneath each contributing author's name to add specific details on the author's contribution.

More information is available in our guide to authors.
<https://www.embopress.org/page/journal/14602075/authorguide>

>> Provide a completed Author Checklist.

>> Adjust the title of the 'Declaration of Interests' section to 'Disclosure and Competing Interests Statement'.

>> Section order should be corrected as follows: Title page - Abstract & Keywords - Introduction - Results - Discussion - Methods - Data Availability - Acknowledgements - Disclosure and Competing Interests Statement - References - Figure Legends - Table(s) - Expanded View Figure Legends.

>> Add a Reagents and Tools table to the Methods section, as a separate file using the existing template in the Guide For Authors, listing key reagents, experimental models, software and relevant equipment.

>> Figure callouts: Ensure that Figure panels Fig 1R, Fig 4D, Suppl. Fig 3B, are called out in the main text.

>> Dataset EV legends: Tables S1 and S2 should be renamed Table EV1 and Table EV2, and both need a legend added to the file, to the top of the page.

>> Please recheck references for the bioRxiv entry Hallgren et al. (2022) and update the citation if in the meantime published as regular article.

>> Data availability section: move the description of the PRIDE dataset to the Data availability section and ensure privacy is released and the data are made publicly accessible.

>> Figures in separate files: Main figures and EV figures should be uploaded as individual, high resolution figure files. Please correct the nomenclature of the supplementary figures to "Figure EV1" etc. Suppl. Fig 1 and 5 span two pages, this is not allowed, please split them into two separate figures.

>> Funding: Please enter the following funding information into our online system: 'the Horizon Europe funding programme under the Marie Skłodowska-Curie Actions Doctoral Networks grant agreement "Explainable AI for Molecules - AiChemist", no. 101120466'.

>> At EMBO Press we ask authors to provide source data for the main manuscript figures. You will receive a separate email with instructions for providing source data with your revised manuscript, including how to upload and organize the files.

Additional information on source data and instruction on how to label the files are available

>> Consider additional changes and comments from our production team as indicated below:

- Figure legends:

1. Please note that the exact p values are not provided in the legends of figures 1A, H; 2M, N
2. Please note that information related to n is missing in the legend of figure 4C
3. Please note that the scale bar needs to be defined for figures 1G, 2E, 5G,
4. Please note that scale bar and its definition are missing for figure 1R

Referee #2:

Comments EMBOJ-2024-119521R-Q revision:

We consider that the authors have addressed satisfactorily all the questions that we raised. We consider that now the manuscript is suitable to be published in EMBO Journal.

Referee #3:

The authors have comprehensively addressed my questions and I have no further concerns. I applaud the ambitious efforts.

The authors addressed the editorial issues.

Dear Dr. Ussar,

Thank you for submitting the revised version of your manuscript. I have now evaluated your amended manuscript and concluded that the remaining minor concerns have been sufficiently addressed.

I am thus pleased to inform you that your manuscript has been accepted for publication in the EMBO Journal.

Related, I would like to hereby ask your consent on keeping the referee response figures included in this file.

On a different note, I would like to alert you that EMBO Press offers a format for a video-synopsis of work published with us, which essentially is a short, author-generated film explaining the core findings in hand drawings, and, as we believe, can be very useful to increase visibility of the work. Please see the following link for representative examples and their integration into the article web page:

<https://www.embopress.org/doi/full/10.15252/emj.2019103932>

Best regards,

Daniel Klimmeck

Daniel Klimmeck, PhD
Senior Editor
The EMBO Journal
EMBO
Postfach 1022-40
Meyerohofstrasse 1
D-69117 Heidelberg
contact@embojournal.org